# Task Vector Bases: A Unified and Scalable Framework for Compressed Task Arithmetic

## Abstract

Task arithmetic, representing downstream tasks through linear operations on task vectors, has emerged as a simple yet powerful paradigm for transferring knowledge across diverse settings. However, maintaining a large collection of task vectors introduces scalability challenges in both storage and computation. We propose Task Vector Bases, a framework compressing $T$ task vectors into $M < T$ basis vectors while preserving the functionality of task arithmetic. By representing each task vector as a structured linear combination of basis atoms, our approach supports standard operations such as addition, negation, as well as more advanced arithmetic ones. The framework is orthogonal to other efficiency-oriented improvements in task arithmetic and can be used in combination with them. We provide theoretical analysis showing that basis compression retains addition generalization guarantees and enables principled unlearning, with error bounds depending on reconstruction quality. Empirically, our proposed basis construction methods consistently outperform heuristic basis construction baselines and, in some cases, even surpass the performance of full task vector collections across diverse downstream applications while reducing storage and computational requirements.

## 1 Introduction

Task vectors (Ilharco et al., 2022) have emerged as a lightweight technique for model editing. Given a downstream task of interest, a task vector is constructed by subtracting the pretrained model weights from those of a fine-tuned model, encoding task-specific information as a direction in parameter space. These vectors can be combined through simple arithmetic operations such as addition and negation, enabling flexible capabilities such as multi-task composition, task analogy or domain generalization, and even task unlearning. Because of their simplicity and effectiveness, task vectors have been widely studied and applied in vision (Chen et al., 2025; Zhu et al., 2025; Tian et al., 2025) and language (Zhao et al., 2024; Wang et al., 2024d; Fu et al., 2025) domains.

Despite these successes, an important question remains: *how well do task vector methods scale with the number of tasks $T$?* From the perspective of task addition, it is often considered efficient compared to multi-task learning on the full mixture of data. But practical deployments increasingly involve dozens of tasks, where both computation and memory footprint still scale linearly with $T$. Storing each task vector, which is the same size as the full model weights, can already be huge for LLMs, and even if disk storage is a relatively moderate cost, the primary systems bottleneck still arises during composition: loading 72 fine-tuned ViT-B/32 models simultaneously can require more than 200GB of memory when combination coefficients are learned with gradient-based methods (Huang, 2023; Li et al., 2024; Yang et al., 2024), making large-scale GPU training infeasible.

To address this, layer-wise merging (Yang et al., 2023) has been proposed to improve flexibility and scalability by operating at the level of individual layers, which leads to a significant performance increase. However, due to the memory constraint, these methods require sequential loading and unloading of layer parameters between CPU and GPU, which incurs significant overhead and prevents full utilization of GPU parallelism (He et al., 2025). This makes these methods prohibitively slow in practice, let alone overlooking cross-layer dependencies within tasks. Beyond addition, in negation, to forget any task in a large collection requires access to specific task vectors to be removed, and thus scales poorly with $T$ when vectors must be stored and retrieved individually. This scaling limitation becomes particularly acute when addition and negation are combined, such as

composing many tasks while selectively removing a subset. Finally, as $T$ grows, optimizing addition itself becomes increasingly difficult, often leading to degraded multi-task performance in both offline composition (Ilharco et al., 2022) and continual merging scenarios (Tang et al., 2025).

In light of the limitations when scaling the number of task vectors, we introduce Task Vector Bases, an algorithm that compresses the entire $T$ task vectors into $M$ basis vectors, yielding a novel unified framework that can be directly integrated with existing task-arithmetic applications. Our framework is orthogonal to and compatible with other compression methods too. Our contributions are:

- **Scalability.** We reduce both storage and computation overhead from a factor of $T$, the number of tasks, to $M$ with $M \leq T$ denoting the number of basis vectors, making task vector methods practical in large-scale or resource-constrained settings at minimum loss of downstream performance. With only 50% of the vectors we can already achieve results better than using 100% of the task vectors, and even when reducing $M$ to $25\% \times T$ we still retain up to 97% of the full performance.
- **Unified framework.** Task Vector Bases provide a unified framework broadly compatible with all weight-space vector steering operations, including offline/online addition and negation.
- **Principled construction.** By learning bases aligned with the geometry of task vectors, our approach avoids the inefficiencies of heuristics such as PCA or random selection and consistently achieves stronger downstream results.
- **Theoretical and empirical validation.** We theoretically compare the generalization performance between full task vectors and compressed bases, and empirically validate the benefits of our method across diverse applications.

## 2 PRELIMINARIES

**Problem Setting** Let $\ell : \mathcal{Y} \times \mathcal{Y} \to \mathbb{R}$ be the loss function, and $h : \mathcal{X} \times \Theta \to \mathcal{Y} \subseteq \mathbb{R}$ be the classifier. When the context is clear, we omit some arguments for $\ell$ and $h$. We consider the initial pre-trained model parameter $\theta_0 \in \mathbb{R}^d$, which is fine-tuned on $T$ tasks to yield fine-tuned parameters $\{\theta_1, \ldots, \theta_T\}$ with respect to the loss functions $\{\ell_1, \ldots, \ell_T\}$. For $n_i$ training samples $D_i = \{(x_{i1}, y_{i1}), \ldots, (x_{in_i}, y_{in_i})\}$ drawn from the $i$-th task distribution $\mathcal{D}_i$, we denote the population risk evaluated at $\theta$ as $\mathcal{L}_i(\theta) = \mathbb{E}_{(x,y) \sim \mathcal{D}_i}[\ell_i(h(x, \theta), y)]$.

**Task Arithmetic (TA) and Applications** Given a collection of $T$ tasks, task vectors are defined as $\tau_i := \theta_i - \theta_0, \forall i \in [T]$, where $\theta_0$ is the pretrained initialization and $\theta_i$ is the fine-tuned model on task $i$. Ilharco et al. (2022) showed that meaningful model behaviors can be obtained through simple arithmetic on task vectors. In general, we view **Offline Task Addition** as producing a merged model

$$\theta_{\text{Add}}^T = \theta_0 + \mathcal{M}(\tau_1, \ldots, \tau_T), \tag{1}$$

where the algorithm $\mathcal{M}$ specifies how the task vectors are combined. Depending on the downstream application, $\mathcal{M}$'s input can be either learned from in-domain data for multi-task learning or from out-of-domain (OOD) validation data for domain generalization. **Online Task Addition** can be applied in a continual setting where $T$ tasks arrive over time. If unlimited storage were available, one could save all past task data exemplars and task vectors (Coleman et al., 2024; Marczak et al., 2024; Chitale et al., 2023), reducing $t$-th step model to be $\theta^{(t)} := \theta_{\text{Add}}^t$. To forget a particular task $j$, we subtract the task vector from $\theta_0$ scaled by a task-specific coefficient $\alpha$, yielding **Task Negation**

$$\theta_{\text{Neg},j} = \theta_0 - \alpha\tau_j, \quad \forall j \in [T]. \tag{2}$$

## 3 IMPROVING EFFICIENCY WITH BASES ARITHMETIC

Under limited compute, it is impractical to save all task vectors for a large number of tasks $T$, and simply impossible when $T \to \infty$ in the online sequential setting. In this study, we focus on the unique computational bottlenecks of task vector methods that arise from operations that scale linearly with $T$. Let space complexity refer to *persistent storage of parameters required to support any future arithmetic operations* (i.e., artifacts that must live on memory and/or disk in the long term), and time complexity refers to the number of *elementary operations needed to produce an edited model prior to inference* from the stored artifacts. Under the setting of Ilharco et al. (2022), supporting task arithmetic operation requires a space complexity of $O(Td)$, since supporting task

negation for any of the possibly randomly selected $T$ tasks means every $\tau_j$ must remain accessible. In terms of time complexity, task addition requires scanning and weighting all $T$ vectors during merging, which costs $O(Td)$ as well. Thus the naive method couples linear space in $T$ with linear-time merging, even if inference time remains $O(d)$ once the edited model is formed.

We propose the framework of **Task Vector Bases**, compressing original $T$ task vectors into $M$ $d$-dim basis vectors $\{B_1, \ldots, B_M\}$ with $M < T$. Bases arithmetic framework can be written as

$$\theta_{\text{Add}}^M = \theta_0 + \mathcal{M}(B_1, \ldots, B_M), \qquad \theta_{\text{Neg},j} = \theta_0 - \alpha \cdot \hat{\tau}_j(B_1, \ldots, B_M), \ \forall j \in [T], \qquad (3)$$

where bases addition replaces any $\tau_i$ with $B_i$, and negation for any of $T$ tasks can be recovered from saved bases. Our goal is to create a compact representation that preserves the information needed to support all existing task arithmetic operations and follow-up improvements built upon naive task arithmetic, while reducing both storage and time complexity from dependence on $T$ to $M$.

### 3.1 PRINCIPLE COMPONENTS AS BASES

A natural candidate for compressing $T$ task vectors into $M < T$ directions is Principal Component Analysis (**PCA**). Let $\mathbf{T} = [\tau_1, \ldots, \tau_T] \in \mathbb{R}^{d \times T}$ denote the task vector matrix stacking task vectors with mean $\mu \in \mathbb{R}^d$. PCA yields $\mathbf{T} - \mu \mathbf{1}^\top \approx (\mathbf{U}_M \mathbf{S}_M) \mathbf{V}_M^\top$, where $\mathbf{B} = \mathbf{U}_M \mathbf{S}_M \in \mathbb{R}^{d \times M}$ serves as the scaled basis matrix and $\mathbf{C} = \mathbf{V}_M^\top \in \mathbb{R}^{M \times T}$ are task-specific coefficients. Original task vector matrix is reconstructed by $\hat{\mathbf{T}} = \mu \mathbf{1}^\top + \mathbf{BC}$. PCA representation requires storing $O(Md)$ basis vectors for addition and additional $T \times M$ coefficients for negation, and all $T$ task vectors can be recovered using transient working memory, thus matching the desired complexity profile and achieving the optimal rank-$M$ approximation in Frobenius norm by Eckart & Young (1936).

Despite its appeal for negation, it is not compatible with the way task addition is typically performed:

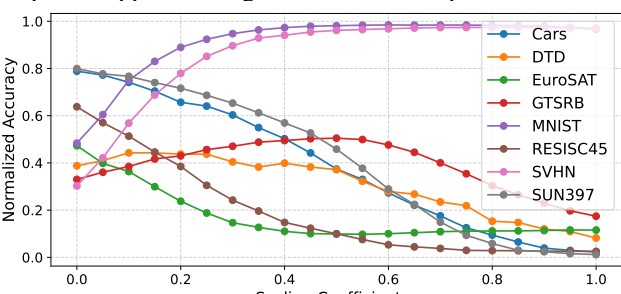
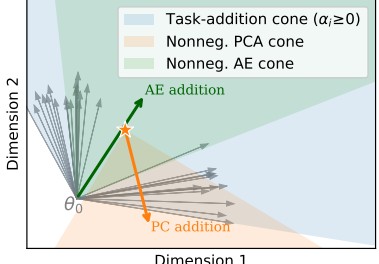

(a) Addition under PCA. Normalized accuracy vs. scaling coefficient $\alpha$ for different datasets using PCA components.

(b) Comparing geometry of task vectors, PC, and our AE bases for addition. Grey arrows are original task vectors from $\mathbf{T}$.

Figure 1: Limitations of PCA for task addition: (a) performance view and (b) geometric view.

1. **Coefficient tuning.** In many addition formulations (Ilharco et al., 2022; Yadav et al., 2024; Ortiz-Jimenez et al., 2024), when applied to task vectors, the merged model can be written as $\theta_{\text{Add}}^T = \theta_0 + \alpha \mathcal{M}(\tau_1, \ldots, \tau_T)$ where $\alpha$ is a single *nonnegative* scalar tuned on validation data shared across task vectors. This formulation assumes that each task vector can be only rescaled positively according to the naming of task *addition*, but PCA bases are arbitrary orthogonal directions with their signs having no semantic meaning. Therefore, these types of addition methods are not compatible with PCA basis: in the left panel of Fig. 1a, half of the datasets cannot even recover the pretrained model accuracy at $\alpha = 0$ by applying Ilharco et al. (2022) on PCA basis. This indicates that these basis components are completely misaligned with the original task vectors, so tuning nonnegative scaling factors cannot interpolate back to the pretrained initialization. Geometrically (right panel of Fig. 1b), this failure arises because first, the bases are not anchored at $\theta_0$ but at ★; second, even if we shift the anchor, the orientation of principal components is determined by the SVD implementation and the sign of PCs (boundary directions of the orange cone) doesn't affect the approximation optimality. The shaded orange cone spanned by nonnegative coefficients does not fully overlap with blue cone from merging original task vectors. In Fig. 1b, tasks to the far left are completely misaligned from PC bases interpolation, leading to irreversible information loss for addition.

2. **Interpretability of directions.** In task arithmetic, each task dataset $D_i$ corresponds to a specific task vector $\tau_i$, so merging methods often rely on this 1-to-1 correspondence. A representative example is Localize-and-Stitch (**L&S**) (He et al., 2024), which trains a binary mask to identify the most

relevant parameters for each task by solving

$$S_i = \arg\min_{S \in \mathbb{R}^d} \mathcal{L}_i\big(\theta_0 + \sigma(S) \odot \tau_i\big) + \lambda \|\sigma(S)\|_1, \tag{4}$$

where the empirical loss is computed on the validation data $D_i$. This method is attractive since the task vectors masked by sparse $S_i$ can be stored at much lower cost than full task vectors, while still enabling accurate model merging. Under a PCA basis representation, assume $\mu = 0$, then bases $\mathbf{B} = \mathbf{U}_M \mathbf{S}_M \approx \mathbf{T} \mathbf{V}_M$ can be written as the linear combination of input vectors, but $\mathbf{V}_M$ involves negative values. This breaks the dataset–vector interpretability: one cannot straightforwardly construct a validation dataset for a PCA basis to learn its mask since negative coefficients would correspond to a nonsensical negative task dataset contribution. We defer more examples to Sec. C.

### 3.2 SOFTMAX-ACTIVATED LINEAR AUTOENCODER AS BASES

#### 3.2.1 BASIS CONSTRUCTION

So how to design bases to preserve the spectral optimality of PCA while aligning the basis direction with original task vectors? We propose to use a softmax activated linear autoencoder, which ensures each basis vector can be interpreted as a convex combination of input task vectors, which is essential for supporting both addition and negation operations in a unified basis framework.

**Definition 3.1** (Autoencoder with softmax encoder and linear decoder). Given parameters $\mathbf{A} \in \mathbb{R}^{T \times M}$, let the encoder weight be a column-wise softmax at temperature $\tau > 0$: $\mathbf{W}_e[:, m] := \mathrm{softmax}\big(\mathbf{A}_{:,m}/\tau\big) \in \Delta^{T-1}, \forall m \in [M]$ and $\mathbf{W}_d \in \mathbb{R}^{M \times T}$ be a linear decoder. We minimize the reconstruction loss as the squared error in Frobenius norm:

$$\mathcal{L}_{\mathrm{AE}}(\mathbf{W}_e, \mathbf{W}_d) = \|\hat{\mathbf{T}} - \mathbf{T}\|_F^2 := \|\mathbf{T}\mathbf{W}_e\mathbf{W}_d - \mathbf{T}\|_F^2. \tag{5}$$

**Lemma 3.2** (Equivalent Gram reformulation). *With gram matrix* $\mathbf{G} := \mathbf{T}^\top \mathbf{T}$ *and* $\mathbf{E} = \mathbf{W}_e\mathbf{W}_d - \mathbf{I}_T$ *as above, Eq.* (5) *is equivalent to*

$$\mathcal{L}_{\mathrm{AE}}(\mathbf{W}_e, \mathbf{W}_d) = \|\mathbf{G}^{1/2}\mathbf{E}\|_F^2 = \mathrm{Tr}(\mathbf{E}^\top \mathbf{G}\mathbf{E}). \tag{6}$$

*Proof.* $\|\mathbf{T}\mathbf{W}_e\mathbf{W}_d - \mathbf{T}\|_F^2 = \|\mathbf{T}\mathbf{E}\|_F^2 = \mathrm{Tr}((\mathbf{T}\mathbf{E})^\top(\mathbf{T}\mathbf{E})) = \mathrm{Tr}(\mathbf{E}^\top \mathbf{T}^\top \mathbf{T}\mathbf{E}) = \mathrm{Tr}(\mathbf{E}^\top \mathbf{G}\mathbf{E})$. Since $\mathbf{G} \succeq 0$, $\mathbf{G}^{1/2}$ is its PSD square root so $\mathrm{Tr}(\mathbf{E}^\top \mathbf{G}\mathbf{E}) = \|\mathbf{G}^{1/2}\mathbf{E}\|_F^2$. □

*Remark* 3.3. The formulation in Eq. (5) requires storing $\mathbf{T} \in \mathbb{R}^{d \times T}$, which scales with model parameters $d$. The Gram reformulation Eq. (6) only depends on $\mathbf{G}, \mathbf{E} \in \mathbb{R}^{T \times T}$, eliminating the $d$-dependence during gradient-based optimization on GPU if precomputing the Gram matrix on CPU.

It is well known that the global optimum of a linear autoencoder without softmax activations can be characterized by PCA solution (Baldi & Hornik, 1989). For any $M < T$, the best rank-$M$ reconstruction's error is given by the spectral bound in both Frobenius and spectral norm:

$$\min_{\mathrm{rank}(\hat{\mathbf{T}}) \leq M} \|\hat{\mathbf{T}} - \mathbf{T}\|_F^2 = \sum_{i=M+1}^{r} \lambda_i(\mathbf{G}), \qquad \min_{\mathrm{rank}(\hat{\mathbf{T}}) \leq M} \|\hat{\mathbf{T}} - \mathbf{T}\|_2^2 = \lambda_{M+1}(\mathbf{G}), \tag{7}$$

where $r = \mathrm{rank}(\mathbf{T})$ and $\lambda_i(\mathbf{G})$ are the eigenvalues of $\mathbf{G}$ in nonincreasing order, with minimum achieved when choosing any $\mathbf{W}_e$ whose column space equals the top-$M$ eigenspace of $\mathbf{G}$ denoted by $S_\star$, and $\mathbf{W}_d$ be the ordinary least square solution given fixed $\mathbf{W}_e$. When softmax activation is applied to the encoder, the loss is at least Eq. (7) with equality conditions below,

**Theorem 3.4** (Exact Achievability with Softmax Encoder). *Using a softmax-activated encoder* $\mathbf{W}_e$, *Eq.* (5) *attains the spectral optimum in Eq.* (7) *if and only if there exist* $M$ *linearly independent vectors* $x_1, \ldots, x_M \in S_\star$ *with strictly positive coordinates.*

#### 3.2.2 BASIS ARITHMETIC

After solving Eq. (6), we define the Autoencoder (**AE**) bases as

$$\mathbf{B} := \mathbf{T}\mathbf{W}_e \in \mathbb{R}^{d \times M} = \left[ \sum_{i=1}^{T} \mathbf{W}_e[i, 1]\tau_i \,\middle|\, \ldots \,\middle|\, \sum_{i=1}^{T} \mathbf{W}_e[i, M]\tau_i \right], \tag{8}$$

where the bases are the convex combinations of original task vectors.

For **offline bases addition**, with $M$ bases, no matter what merging method $\mathcal{M}$ we use, we always subsample $n_i \cdot M/T$ validation data from each task dataset $D_i$, and re-use all existing merging

methods directly on $M$ bases paired with subsampled dataset denoted as $\widetilde{D}_i$. This immediately reduces the time complexity of data-based merging methods from $\times T$ evaluations to $\times M$, since they are now applied on $Mn_i$ effective data points in $\cup_{i=1}^T \widetilde{D}_i$.

To see how AE bases solving the limitations of PCs, for coefficient tuning methods, softmax bases guarantee that each $B_m$ is a convex combination of task vectors. If we apply Ilharco et al. (2022) for addition, any nonnegative mixture of the bases $\sum_{m=1}^M \alpha_m B_m$, $\alpha_m \geq 0$ remains a nonnegative linear combination of the original $\tau_i$ by plugging in Eq. (8). Hence the feasible region spanned by softmax bases is always a subset of the nonnegative cone defined by task addition like in Fig. 1b.

Besides, in settings with 1-to-1 correspondence between tasks and datasets, we define validation data mixture for basis $B_m$ with its encoder weights $\mathbf{W}_e[:,m]$, which specify how input tasks contribute to the basis. For example, to use He et al. (2024), for basis $m$, the effective objective becomes

$$S_m = \arg\min_{S \in \mathbb{R}^d} \Big( \sum_{i=1}^T \mathbf{W}_e[i,m] \, \mathcal{L}_i\big(\theta_0 + \sigma(S) \odot B_m\big) \Big) + \lambda \|\sigma(S)\|_1. \tag{9}$$

That is, instead of attaching one dataset to one task vector, each basis $B_m$ is paired with a convex combination of the original task validation losses, weighted by the encoder weights, thus $S_m$ is now a joint mask applicable to multiple tasks. This allows Localize-and-Stitch to remain applicable in the basis setting while requiring space only for the $M$ bases rather than all $T$ task vectors to reduce the memory burden. See Tab. 5 for more bases addition examples with existing merging algorithms.

---

**Algorithm 1** Online Bases Addition

**Require:** Buffer budget $M$, basis construction pipeline AE_TRAIN($\cdot$)
1: Initialize basis set $\mathbf{B} \leftarrow \emptyset$ and $k \leftarrow 0$
2: **for** each new task $t = 1, 2, \dots$ **do**
3:      Receive new task vector $\tau_t \in \mathbb{R}^d$
4:      **if** $k < M$ **then**
5:          $\mathbf{B} \leftarrow \mathbf{B} \cup \{\tau_t\}, k \leftarrow k + 1$
6:      **else**
7:          $(\mathbf{W}_e, \mathbf{W}_d) \leftarrow$ AE_TRAIN$(\mathbf{B}, M)$
8:          $\mathbf{U} = \mathbf{B}\mathbf{W}_e \in \mathbb{R}^{d \times (M-1)}$
9:          $\mathbf{B} \leftarrow [\mathbf{U}, \tau_t] \in \mathbb{R}^{d \times M}$
10:      $\theta^{(t)} = \theta_0 + \mathcal{M}_{\cup_{i=1}^T \widetilde{D}_i}(\mathbf{B})$

---

For **online bases addition**, we consider a more practical limited-compute setting where only $M$ finite vectors can be stored persistently in Alg. 1. At step $t$, when we found the buffer is full, we apply the autoencoder compression to reduce these $M$ vectors back into $M-1$ bases, then we put a new task vector $\tau_t$ also back into the buffer, ensuring storage cost remains fixed as $O(Md)$ while supporting an unbounded sequence of tasks. Besides, per step time complexity is also only depend on $M$: the compression to reduce $M$ vectors down to $M-1$ cost $O(M^2 d)$, and the merging algorithm operates on $M$ bases. Note that although we call the AE training pipeline for each step $t$, empirically compared to line 10's addition step, the basis processing step cost is negligible (Sec. E.2).

For **bases negation**, we first reconstruct the full task vector matrix

$$\hat{\mathbf{T}} = \mathbf{B}\mathbf{W}_d, \tag{10}$$

and then apply negation directly on the reconstructed vectors so that we forget task $j$ via $\theta_0 - \alpha\hat{\tau}_j = \theta_0 - \alpha\hat{\mathbf{T}}[:,j]$. In terms of storage, we only need to persist the $M \times d$ bases $\mathbf{B}$ and the decoder weight $\mathbf{W}_d \in \mathbb{R}^{M \times T}$, but since $M < T \ll d$, the total storage is still much smaller than the naive $O(Td)$ required to store all task vectors directly, and thus matches our efficiency goal.

### 3.2.3 THEORETICAL GUARANTEES

We briefly state our theoretical analysis guided by Taylor expansion for standard task arithmetic in Ilharco et al. (2022) and their counterparts under basis representations. See proof details in Sec. B.2.

**Theorem 3.5** (Task Addition & Basis Addition). *For $\theta_{\text{Add}}^T = \theta_0 + \sum_{i=1}^T \alpha_i \tau_i$, if we introduce constants including task vector norm upper bound $C$, loss smoothness $L_i$, task vector similarity $\epsilon$, $\forall i \in [T]$, the generalization gap between the merged model and finetuned model is bounded by:*

$$\mathcal{L}_i(\theta_{\text{Add}}^T) - \mathcal{L}_i(\theta_i) \leq L_i C(1 + \epsilon). \tag{11}$$

*When each basis is defined in Eq. (8), and bases addition merged model is $\theta_{\text{Add}}^M = \theta_0 + \sum_{m=1}^M \alpha_m B_m$, the same bound in Eq. (11) holds.*

**Theorem 3.6** (OOD Generalization with Task Vectors & Bases). *Suppose $\tau_{\text{tar}}$ is an unseen task vector and we want to generalize to this target task with Eq. (1) with existing task vectors in $\mathbf{T}$. If $\exists i^\star \in [T]$ with $\langle \tau_{\text{tar}}, \tau_{i^\star} \rangle \geq \gamma C$, then there exists set of merging coefficients $\alpha_i, \forall i \in [T]$ such that*

$$\mathcal{L}_{\text{tar}}(\theta_{\text{Add}}^T) \ \leq \ \mathcal{L}_{\text{tar}}(\theta_{\text{tar}}) + L_{\text{tar}}C(1 - \gamma). \tag{12}$$

*If we use basis instead when $\theta_{\text{Add}}^M = \theta_0 + \sum_{m=1}^{M} \alpha_m B_m$, if some basis $B_m$ contains $\tau_{i^\star}$ with weight at least $\rho$, i.e. $\mathbf{W}_e[i^\star, m] \geq \rho$,*

$$\mathcal{L}_{\text{tar}}(\theta_{\text{Add}}^M) \ \leq \ \mathcal{L}_{\text{tar}}(\theta_{\text{tar}}) + L_{\text{tar}}C(1 - \rho\gamma). \tag{13}$$

**Theorem 3.7** (Task Negation & Basis Negation). *For task vector negation $\theta_{\text{Neg},i} = \theta_0 - \alpha_i\tau_i$, for all control tasks $j \neq i$, the performance gap compared to pretrained model is bounded by:*

$$\mathcal{L}_j(\theta_{\text{Neg},i}) - \mathcal{L}_j(\theta_0) \ \leq \ L_j C\left(\tfrac{3}{2} + \epsilon\right). \tag{14}$$

*Let $\widehat{\tau}_i$ be the $i$-th reconstructed task vector from Eq. (10). If Eq. (5) is minimized to the spectral lower bound, for $\theta_{\text{Neg},i} = \theta_0 - \alpha_i\widehat{\tau}_i$:*

$$\mathcal{L}_j(\theta_{\text{Neg},i}) - \mathcal{L}_j(\theta_0) \ \leq \ L_j C\left(\frac{5}{2} + 2\epsilon\right) + L_j\lambda_{M+1}(\mathbf{G}). \tag{15}$$

*Remark* 3.8. These claims establish the guarantees for bases arithmetic extend naturally from the original task vector setting. In the case of addition (Thm. 3.5, Thm. 3.6), the bounds remain structurally identical: replacing $\tau_i$ with bases $B_m$ incurs no extra penalty, except that the alignment constant $\gamma$ is weakened by at most the encoder coverage factor $\rho$. For task negation (Thm. 3.7), one key difference is that basis reconstruction introduces a term controlled by the residual, i.e. $\lambda_{M+1}(\mathbf{G})$, which vanishes when $M$ is large enough to capture the principal components of $\mathbf{G}$.

## 4 EXPERIMENTS

We present the experiments organized by task arithmetic application under basis framework. Details of datasets, metrics, hyperparameters, and additional experiments including verification of theoretical claims (Sec. D), sensitivity of $\tau$ (Sec. E.1), choice of subsampling/weighting (Sec. F.1), results on generative tasks (Sec. F.2) are deferred to Appendix.

### 4.1 BASES ADDITION

#### 4.1.1 OFFLINE MULTITASK LEARNING

Table 1: Comparison of absolute addition accuracy across ViT models under 8, 14, and 20 vision tasks (Wang et al., 2024a) with $M = 50\%$ of total tasks. Bold entries are the best-performing basis method within each block, while underlined entries are cases where basis addition outperforms full task-vector addition. See normalized accuracies and per dataset results in Tab. 13, and Figs. 9 to 11.

| Method | ViT-B/16 | | | ViT-B/32 | | | ViT-L/14 | | |
|---|---|---|---|---|---|---|---|---|---|
| | 8 task | 14 task | 20 task | 8 task | 14 task | 20 task | 8 task | 14 task | 20 task |
| Pretrained | 0.554 | 0.620 | 0.598 | 0.481 | 0.569 | 0.556 | 0.698 | 0.691 | 0.656 |
| Finetuned | 0.924 | 0.913 | 0.916 | 0.904 | 0.893 | 0.898 | 0.943 | 0.934 | 0.935 |
| TA (Ilharco et al., 2022) | 0.754 | 0.705 | 0.658 | 0.708 | 0.653 | 0.605 | 0.850 | 0.794 | 0.740 |
| RandSelect | 0.645 | 0.649 | 0.620 | 0.643 | 0.638 | 0.611 | 0.697 | 0.727 | 0.609 |
| PCA | 0.495 | 0.578 | 0.573 | 0.532 | 0.571 | 0.585 | 0.589 | 0.653 | 0.642 |
| AE (Ours) | **0.666** | **0.673** | **0.635** | **0.689** | **0.660** | **0.613** | **0.736** | **0.753** | **0.715** |
| TIES (Yadav et al., 2024) | 0.797 | 0.732 | 0.682 | 0.751 | 0.680 | 0.634 | 0.869 | 0.795 | 0.757 |
| RandSelect | 0.664 | 0.659 | 0.620 | 0.655 | 0.649 | 0.627 | 0.733 | 0.733 | 0.708 |
| PCA | 0.496 | 0.578 | 0.573 | 0.533 | 0.571 | 0.595 | 0.589 | 0.652 | 0.644 |
| AE (Ours) | **0.672** | **0.672** | **0.635** | **0.687** | **0.651** | 0.607 | **0.742** | **0.754** | **0.711** |
| L&S (He et al., 2024) | 0.759 | 0.681 | 0.601 | 0.767 | 0.652 | 0.598 | 0.778 | 0.753 | 0.701 |
| RandSelect | 0.553 | 0.534 | 0.437 | 0.546 | 0.494 | 0.434 | 0.670 | 0.677 | 0.601 |
| PCA | 0.523 | 0.450 | 0.410 | 0.467 | 0.408 | 0.361 | 0.667 | 0.589 | 0.543 |
| AE (Ours) | **0.667** | **0.672** | **0.641** | **0.691** | **0.667** | **0.628** | **0.736** | **0.732** | **0.729** |

Tab. 1 compares basis construction strategies across ViT models for vision tasks, and Tab. 2 is the comparison on the language benchmark with RoBERTa models. We include 3 popular merging methods, TA, TIES with coefficient tuning, and L&S where the last one can be used to additionally

Table 2: Comparison of absolute addition accuracy with RoBERTa-base model on 12 language task benchmark with bases number $M = 25\%$ of the total tasks. 100% means using all task vectors for corresponding merging methods. See the normalized accuracy version and full per dataset results in Tab. 14 and Fig. 12. With 25% of task vectors, we can recover up to 97% (L&S-AE) of the accuracy.

| **TA** (Ilharco et al., 2022) | | | | **TIES** (Yadav et al., 2024) | | | | **L&S** (He et al., 2024) | | | |
|---|---|---|---|---|---|---|---|---|---|---|---|
| 100% | RandSelect | PCA | AE | 100% | RandSelect | PCA | AE | 100% | RandSelect | PCA | AE |
| 0.626 | 0.453 | 0.449 | **0.472** | 0.600 | 0.453 | 0.469 | **0.470** | 0.759 | 0.619 | 0.623 | **0.733** |

compress task vectors with sparsity, and compare three reduced-basis approaches: RandSelect (randomly selecting available tasks), PCA, and our AE. We keep 50% of the vectors in bases for vision and 25% for languauge experiments. For a fair comparison, all bases methods use the same sub-sampling strategy in Sec. 3.2.2 to only use $n_i M/T$ validation data. While constructing L&S bases, in RandSelect, we allow the method to only learn task masks for the selected task vectors, and in PCA, since we cannot disentangle nonnegative contributions from each original task to a principal component, we assign uniform weights across tasks where $\mathbf{W}_e[i,m] = 1/T$ in Eq. (9). See the alternative baseline only using positive weights for PCA in Tab. 13.

Across nearly all settings, AE achieves the best performance within each method block, consistently outperforming both RandSelect and PCA, implying that learning a compact AE basis captures more useful task interactions than other methods. The advantage of AE is especially pronounced in L&S, where interpretability of bases vectors is central to the method's validity. For several large-task regimes, AE or even RandSelect can outperform full-task merging, showing that fewer but more coherent vectors may lead to better generalization while reducing storage cost and merging time. As predicted in Sec. 3.1, PCA consistently performs poorly in vision experiments, and sometimes even worse than the pretrained baseline. We leave the comparison of bases methods across $M$ in Fig. 8.

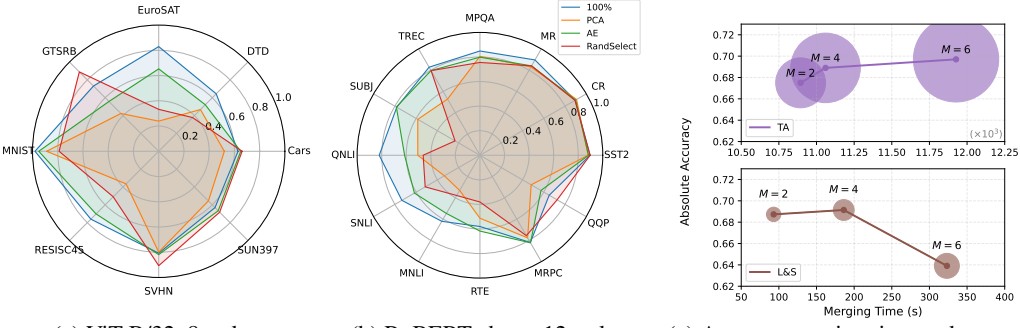

(a) ViT-B/32, 8 tasks.   (b) RoBERTa-base, 12 tasks   (c) Acc. vs. merging time and storage.

Figure 2: (a)–(b) Radar plots showing per-task accuracy across vision (100% = TA) and language (100% = L&S) benchmarks. (c) Absolute accuracy against merging time for different $M$, with circle size indicating disk storage cost in gigabytes (same scale across top and bottom).

For per task results in Figs. 2a and 2b, we observe a nested pattern where the 100% merge generally dominates or on par with AE, and AE in turn dominates PCA. Therefore, using all available task vectors provides the strongest signal, AE compresses them while preserving most of the structure, and PCA mixes wrong directions, leading to degraded performance. Unlike AE and PCA, Rand-Select is inherently unstable: by dropping more than half the task vectors blindly, it can achieve remarkably strong performance on particular tasks (e.g., GTSRB and SVHN), but this comes at the cost of severe degradation on other tasks where critical information is lost (e.g., DTD and SUBJ).

Fig. 2c show accuracy versus merging time, with bubble size indicating storage cost. Clearly, both merging time and storage grow with the number of bases $M$. This highlights that our basis compression method provides improvements in both time and space efficiency due to $M < T$. Importantly, our basis compression is complementary to sparsity-based approaches like L&S, showing compatibility with existing task vector compression frameworks which may further compress bases storage up to roughly 90% if bases are saved in CSR format. In terms of accuracy, for TA, increasing $M$ yields better accuracy but for L&S, however, accuracy does not monotonically improve with larger $M$. In fact, accuracy drops for $M = 100\%$ in Tab. 1 for certain settings, and adding too many task vectors may actually hurt performance due to increasing task conflicts Ilharco et al. (2022).

| Method | 2 shot | 16 shot |
|---|---|---|
| aTLAS (Zhang et al., 2024) | 0.826 | 0.837 |
| aTLAS$_{subsample}$ | 0.819 | 0.835 |
| RandSelect | 0.821 | 0.829 |
| PCA | 0.817 | 0.829 |
| AE (Ours) | **0.822** | **0.830** |
| aTLAS$^{\geq 0}$ | 0.825 | 0.833 |
| aTLAS$^{\geq 0}_{subsample}$ | 0.820 | 0.830 |
| RandSelect | **0.821** | 0.827 |
| PCA | 0.816 | 0.822 |
| AE (Ours) | 0.820 | **0.828** |

Table 3: ViT-B/32 results with OOD 6 tasks at $M = 50\%$ of in domain 8 tasks.

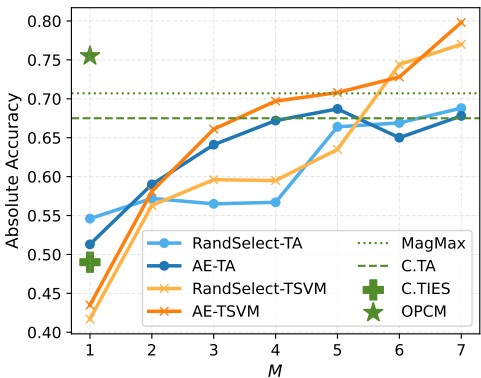

Figure 3: Online continual results on ViT-B/32 with 8 tasks varying the size of storage buffer.

### 4.1.2 OFFLINE FEWSHOT OOD GENERALIZATION

Tab. 3 presents addition evaluated on unseen 6 OOD tasks by merging in domain 8 task vectors from Tab. 1 under a few shot setting where direct finetuning on tiny target subset only creates weak models. We use aTLAS (Zhang et al., 2024) as the base addition method, a flexible framework where each scaling coefficient is a learned block matrix (more details see Tab. 5). We compare its standard formulation with a square-parameterized version aTLAS$^{\geq 0}$, which guarantees the nonnegativity of coefficients simulating (Ilharco et al., 2022) at the cost of minor performance drop.

We benchmark aTLAS and aTLAS$_{subsample}$ (trained with $100\%$ task vectors but only $50\%$ of the coefficient-learning data, and basis methods with $M = 50\%$, evaluated under both unconstrained and nonnegative merging coefficients settings. In Tab. 3 $k$-shot refers to the number of per class samples for aTLAS without any subsampling. Key observations include: first, performance gap among basis methods is smaller than in Tab. 1, but PCA remains consistently worse likely due to its misaligned anchor not centered at $\theta_0$ mentioned in Sec. 3.1. Second, our AE method outperforms RandSelect and PCA in 3 out of 4 settings, particularly when coefficients are unconstrained, showing AE's flexibility. Finally, with very limited data (2-shot), basis methods slightly outperform aTLAS$_{subsample}$ since aTLAS has higher degrees of freedom and requires more data during learning. But in 16-shot, aTLAS$_{subsample}$ catches up and surpasses the bases methods.

### 4.1.3 ONLINE CONTINUAL LEARNING

Fig. 3 illustrates the online continual task merging setting where we fix $M$ checkpoints stored in persistent memory and evaluate the final merged model $\theta^{(t)}$ on all $t$ tasks seen in the sequence. We compare two basis construction methods, RandSelect and AE, paired with two merging rules: TA and TSVM (Gargiulo et al., 2025). Note that the full offline TSVM accuracy with $M = 100\%$ is 0.857. PCA is omitted since it consistently underperforms in offline addition experiments. For baselines, we also include prior continual merging methods in green: MagMax (Marczak et al., 2024) (selecting maximum-magnitude task vector entries), Continual TA $\theta^{(t)} = \theta^{(t-1)} + \lambda \tau_t = \theta_0 + \lambda \sum_{i=1}^{t-1} \tau_i + \lambda \tau_t$, Continual TIES, and OPCM (Tang et al., 2025). In prior work, continual merging was only defined for $M = 1$, i.e., storing a single model checkpoint $\theta^{(t-1)}$ and merging it with the new task vector $\tau_t$. However, methods like MagMax and Continual TA can be reformulated as running statistics, making their $M = 1$ version equivalent to the offline $M = 100\%$ limit.

We see that as $M$ increases, accuracy steadily improves, and AE almost consistently outperforms RandSelect across both TA and TSVM, showing clear advantages for most values of $M$, although when $M \to T$, AE and RandSelect roughly coincide as they are both approaching full-rank approximation. Comparing to a fixed green baseline method, AE achieves better performance than RandSelect with fewer checkpoints. For example, with $M = 4$ (50%), AE–TSVM already surpasses Continual TA, while RandSelect requires $M = 6$ (75%). Finally, although MagMax and OPCM are specifically designed for online continual merging, pairing AE with a strong offline merging method (TSVM here) eventually outperforms specialized baselines once $M$ is moderately large. Thus, even a weak basis method like RandSelect, combined with an effective offline $\mathcal{M}$, provides strong continual merging performance without specialized online setup adaptations. With future advances in the field, we expect even smaller $M$ values to surpass SOTA continual merging baselines. The result across different sizes of ViT is included in Tab. 16 in the Appendix.

## 4.2 BASES NEGATION

In negation experiments, since RandSelect cannot be directly applied to negation (discarded task vectors cannot be recovered or inferred from saved bases without retraining task vectors), we propose RandProj as the random baseline, where bases are defined as the random orthogonal matrix $\mathbf{Q} \in \mathbb{R}^{d \times M}$ obtained from QR decomposition of a Gaussian random matrix. We save projection coefficients $\mathbf{C} = \mathbf{Q}^\top \mathbf{T} \in \mathbb{R}^{M \times T}$, and during negation, we reconstruct $T$ task vectors by $\hat{\mathbf{T}} = \mathbf{QC} = \mathbf{QQ}^\top \mathbf{V}$, projecting each task vector onto the random subspace spanned by $\mathbf{Q}$.

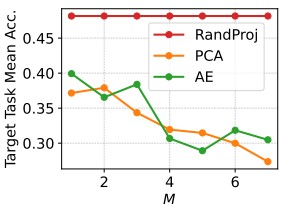

Figure 4: Target task forgetting as a function of $M$.

| Method | ViT-B/16 | | ViT-B/32 | | ViT-L/14 | |
|---|---|---|---|---|---|---|
| | Target ($\downarrow$) | Control | Target ($\downarrow$) | Control | Target ($\downarrow$) | Control |
| TA | 0.213 | 0.654 | 0.240 | 0.649 | 0.190 | 0.729 |
| RandProj | 0.494 | 0.683 | 0.482 | 0.633 | 0.589 | 0.755 |
| PCA | 0.190 | 0.646 | 0.319 | 0.610 | 0.178 | 0.724 |
| AE | 0.255 | 0.659 | 0.307 | 0.605 | 0.270 | 0.734 |

Table 4: Target and control metrics comparison averaged under 8 vision tasks' unlearning setting. Shaded methods use reconstructed task vectors from $M = 50\%$ bases.

Tab. 4 compares methods under the 8-task setting with $M = 50\%$, reporting performance on both target and control tasks. On the target tasks, lower accuracy indicates better forgetting, while on control tasks (ImageNet (Deng et al., 2009)), higher accuracy indicates better retention of pretrained knowledge. We observe that PCA and AE both achieve significant forgetting compared to RandProj, and the difference between PCA and AE metrics can be treated as tradeoffs between target and control metrics. Fig. 4 reports target task mean accuracy as a function of the number of bases $M$. Both PCA and AE gradually comparably reduce target task accuracy as $M$ increases. This behavior is expected from Eq. (7): with suitable hyperparameter tuning, the softmax AE variant can approximate the same spectral lower bound as PCA. In contrast, RandProj remains flat at roughly the same level as the pretrained model $\theta_0$, showing that it fails to forget even as $M$ grows, since random projections do not align with task-specific directions.

## 5 RELATED WORK

A number of recent efforts have sought to make task vector methods more scalable through compression, and we discuss the broader scope of task arithmetic and model merging in Sec. A. One line of work focuses on localization or sparsification, identifying subsets of parameters most relevant for each task and masking out the rest. By sparsifying task updates into different subspaces, these methods reduce task interference and improve memory efficiency, since only sparse weights or masks are stored (He et al., 2024; Yadav et al., 2024; Yu et al., 2024; Davari & Belilovsky, 2025; Tang et al., 2023a; Wang et al., 2024b). A complementary line explores quantization (Liu et al., 2024; Huang et al., 2025), where task vectors are quantized directly without notable degradation in merging performance (Kim et al., 2025). Sparsification and quantization both act along the parameter-dimension axis of the task matrix (reducing $d$), while our Task Vector Bases approach operates on the task-count axis (reducing $T$). For single-task model merging (Sec. A.1), prior work proposes alternative optimization algorithms (Li et al., 2024) or assumes fine-tuned weights lie in a thin Gaussian shell (Jang et al., 2025), a different context from our multitask setup.

## 6 CONCLUSION

We introduced Task Vector Bases, a unifying framework for compressing collections of task vectors into a compact set of basis vectors. This approach addresses the key computational bottlenecks of task vector methods—space and time complexity scaling linearly with the number of tasks—while preserving compatibility with all standard arithmetic operations. Empirically, Task Vector Bases not only reduce storage and computation but also improve task performance over heuristic alternatives such as PCA or random selection. Our analysis further clarifies the generalization performance difference between full task vectors and compressed bases, showing that bases provide a scalable and effective representation for model editing. We hope this work establishes Task Vector Bases as a practical building block for future research on efficient and interpretable weight space interventions.

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

# A ADDITIONAL RELATED WORK

## A.1 SINGLE-TASK MERGING METHODS

Prior to Task Arithmetic (Ilharco et al., 2022), researchers discussed how to combine models fine-tuned on the same task, with some minor differences due to hyperparameter changes, as an alternative to ensembles, starting with model soup (Wortsman et al., 2022). Since fine-tuned models capture more domain-specific skills while pretrained models contain more generic knowledge, WiSE-FT (Wortsman et al., 2021) proposed merging the pretrained model and the fine-tuned model via linear interpolation, achieving balanced or even optimal performance on both in-domain and out-of-distribution generalization metrics. (Izmailov et al., 2018) introduced stochastic weight averaging, which includes intermediate checkpoints before model convergence for model merging. Several close variants, such as exponentially moving averaging (Szegedy et al., 2016) and LAtest Weight Averaging (Kaddour, 2022; Sanyal et al., 2023), have been explained theoretically under a unified framework (Wang et al., 2024c).

## A.2 MULTI-TASK MERGING METHODS

The major difference from Sec. A.1 is that all methods discussed in this subsection focus on the setting that one pretrained model is fine tuned on many different tasks. Task arithmetic (Ilharco et al., 2022) can be seen as the generalization of the single-task model merging method, model soup (Wortsman et al., 2022), where task vectors are simply averaged. In (Ilharco et al., 2022), however, the scaling coefficients $\alpha$ are allowed to be tuned. Since then, several ideas have been proposed to improve task arithmetic. First, since tuning $\alpha$ is time-consuming, popular approaches such as Fisher merging (Matena & Raffel, 2022), RegMean (Jin et al., 2022), AdaMerging (Yang et al., 2023), Evol (Akiba et al., 2024) aim to find better methods to automatically adjust scaling coefficients for improved task arithmetic performance. Second, instead of using standard fine-tuning to obtain $\tau$, alternative fine-tuning methods, such as tangent space fine-tuning (Ortiz-Jimenez et al., 2024) and parameter-efficient fine-tuning methods (Zhang et al., 2023; Tang et al., 2023b; Stoica et al., 2024), are employed in task arithmetic to disentangle task information for better merging. Third, to reduce task vector conflicts, task vectors can be sparsified into different subspaces by localization as we discussed in Sec. 5. Finally, inspired by the Mixture-of-Experts (Shazeer et al., 2017) mechanism, task vector merging performance can be enhanced by learned routers that dynamically merge task-specific and task-shared information (Lu et al., 2024; Tang et al., 2024). For more details on the latest task arithmetic methods and their applications, we refer readers to the model merging survey (Yang et al., 2024).

## A.3 BASES IN LOW-RANK AND SUBSPACE MERGING METHODS

Task Singular Vectors (TSV, -M for merging and -C suffix for compression) (Gargiulo et al., 2025) and Marczak et al. (2025) study task updates at the layer level and apply SVD to decompose unflattened task matrices, where the latter further differentiate between task-shared and task-specific subspaces. While these works also rely on eigenbasis constructions, they differ in both motivation and scope from ours: they both require access to all full task vectors when computing singular components, and they primarily target improving addition performance when $M = 100\%$. Besides, TSV-C only supports model compression only when the task metadata is known or inferred through routing based merging methods (Tang et al., 2024). In contrast, our Task Vector Bases framework is designed as a general compression mechanism that unifies any downstream applications not only limited to addition, and aim to approximate (typically treated as upper bound) $M = 100\%$ metrics with $M < 100\%$ bases.

## A.4 MATRIX FACTORIZATION AND DIMENSIONALITY REDUCTION METHODS

A related line of work has explored nonnegative variants of matrix factorization such as nonnegative PCA (Montanari & Richard, 2015) and nonnegative matrix factorization (NMF) (Févotte & Idier, 2011; Cichocki & Phan, 2009). These approaches have been proposed as remedies for the interpretability limitations of PCA, since enforcing nonnegativity on either the basis vectors or the coefficients ensures that components can be interpreted as additive components. However, applying these methods in our setting is not straightforward. Standard NMF requires the input matrix

itself to be nonnegative, which is incompatible with task vectors that contain signed weight updates. Nonnegative PCA similarly constrains basis vectors to the nonnegative orthant, preventing them from aligning with unconstrained task vector directions. Another family of dimensionality reduction methods includes sparse coding and dictionary learning (Mairal et al., 2009), which learn basis atoms and sparse codes for reconstructing high-dimensional data. While these approaches are applicable to signed inputs, they differ from our design in a critical way. In sparse coding, the learned basis vectors are unconstrained. In summary, even if preprocessing tricks for NMF (e.g., splitting positive and negative channels or affine shifts) are applied, both type of methods distort the geometry of the task-vector cone and break the guarantees needed for task arithmetic: adding coefficient vectors (the operation underlying task addition) may yield mixtures outside the cone spanned by the original tasks as in PCA, thus breaking the structure that our analysis relies upon and can widen the addition generalization gap.

# B PROOF DETAILS

## B.1 EXACT ACHIEVABILITY WITH SOFTMAX ENCODER

**Lemma B.1** (Softmax surjects onto the simplex interior). *Write* $\mathrm{int}(\Delta^{T-1}) = \Delta^{T-1} \cap \mathbb{R}_{++}^T$ *for the interior of the simplex. For any* $b \in \mathrm{int}(\Delta^{T-1})$ *and any* $\tau > 0$, *there exists* $a \in \mathbb{R}^T$ *such that* $\mathrm{softmax}(a/\tau) = b$. *One choice is* $a_i = \tau \log b_i + c$ *for any constant* $c \in \mathbb{R}$.

*Proof.* This is immediate from the definition of softmax and the invariance under adding a constant: $\mathrm{softmax}(z)_i = e^{z_i} / \sum_j e^{z_j}$. $\qquad\square$

**Theorem 3.4** (Exact Achievability with Softmax Encoder). *Using a softmax-activated encoder* $\mathbf{W}_e$, *Eq. (5) attains the spectral optimum in Eq. (7) if and only if there exist* $M$ *linearly independent vectors* $x_1, \ldots, x_M \in S_\star$ *with strictly positive coordinates.*

*Proof.* Let $\mathbb{R}_{++}^T := \{x \in \mathbb{R}^T : x > 0\}$. The statement is equivalent to say $x_j \in \mathbb{R}_{++}^T$ for all $j$.

($\Rightarrow$) If the optimum is achieved by some $\mathbf{W}_e$ with columns $w_1, \ldots, w_M \in \mathrm{int}(\Delta^{T-1})$, then (Baldi & Hornik, 1989) implies that the column space of $\mathbf{W}_e$ must equal $S_\star$. Since each $w_m$ is strictly positive, we conclude $w_m \in S_\star \cap \mathbb{R}_{++}^T$, and the $w_m$ are linearly independent as they span $S_\star$.

($\Leftarrow$) Conversely, suppose there exist $M$ independent $x_1, \ldots, x_M \in S_\star \cap \mathbb{R}_{++}^T$. Normalize each to sum to one, $w_m := x_m/(\mathbf{1}^\top x_m) \in \mathrm{int}(\Delta^{T-1})$. Set $\mathbf{W}_e = [w_1 \ \cdots \ w_M]$, which has column space $S_\star$. By surjectivity of softmax (Lemma B.1), there exists $\mathbf{A}$ such that $\mathrm{softmax}(\mathbf{A}/\tau) = \mathbf{W}_e$ (column-wise softmax). Then by (Baldi & Hornik, 1989) with the least-squares optimal decoder $\mathbf{W}_d$, the reconstruction error equals the spectral bound, achieving the optimum. $\qquad\square$

## B.2 GENERALIZATION OF TASK AND BASES ARITHMETIC

### B.2.1 COMMON ASSUMPTIONS

We first introduce several practical shared common assumptions used in our theorems.

**Assumption B.2** (Fine-tuning Regime). *We assume that* $\forall i \in [T], \frac{\partial \mathcal{L}_i(\theta_i)}{\partial \theta} = \mathbf{0}$ *and* $\exists C > 0$ *such that* $\|\tau_i\|^2 \le C$.

This assumption is often met in practice since $\theta_i$ is fine-tuned from the pre-trained model $\theta_0$ on the particular downstream task $\mathcal{D}_i$ until convergence. Furthermore, during the fine-tuning regime, the change of model parameters is relatively small. Through a sparsity localization technique, (He et al., 2024) show that it is sufficient to only fine-tune 1%~5% of the model parameters for competitive performances.

**Assumption B.3** (Local Smoothness). *Any fine tuning loss function* $\mathcal{L}$ *is* $L_i$-*locally smooth w.r.t. model parameters at* $\theta_i$, *which means for any* $\theta \in \Theta$ *such that* $\|\theta - \theta_i\|^2 = O(C), \mathcal{L}(\theta) - \mathcal{L}(\theta_i) \le \left\langle \theta - \theta_i, \frac{\partial \mathcal{L}(\theta_i)}{\partial \theta} \right\rangle + \frac{L_i}{2} \|\theta - \theta_i\|^2$. *Note that* $\theta_i$ *is the fine-tuned model trained on* $\mathcal{D}_i$ *and* $L_i =$

$\|\mathbf{H}(\theta_i)\|_2$ is the spectral norm of the Hessian matrix of $\mathcal{L}$, evaluated locally at $\theta_i$. We hide the subscript of $L_i$ when the context is clear.

Smoothness is a standard assumption in optimization theory (Garrigos & Gower, 2023) and has been used in recent work on Sharpness-Aware Minimization (Foret et al., 2020; Wen et al., 2022) to encourage flatter minima and improve generalization. Since we focus mainly on the fine-tuning regime in the analysis, we only consider smoothness in a local region.

**Assumption B.4** (Scaling Coefficients). *Let $\alpha_1, \ldots, \alpha_T$ be the coefficients used to scale the task vector in task arithmetic. We assume $\alpha_i \geq 0, \forall i$ and $\sum_{i \in [T]} \alpha_i = 1$.*

### B.2.2 TASK ADDITION & BASIS ADDITION

**Theorem B.5** (Task Addition for Multitask Learning). *Let $0 < \epsilon \leq 1$ be a universal constant such that $\forall i \neq j, |\cos(\tau_i, \tau_j)| \leq \epsilon.$[1] Let task addition $\theta^T_{\mathrm{Add}} = \theta_0 + \sum_{i=1}^{T} \alpha_i \tau_i$ be the model parameter used for multitask learning, then $\forall i \in [T]$,*

$$\mathcal{L}_i(\theta^T_{\mathrm{Add}}) - \mathcal{L}_i(\theta_i) \leq L_i C(1 + \epsilon). \tag{16}$$

*Proof.* Note that since $\theta^T_{\mathrm{Add}} = \theta_0 + \sum_{i=1}^{T} \alpha_i \tau_i$, so

$$\|\theta^T_{\mathrm{Add}} - \theta_0\|^2 = \left\| \sum_{i=1}^{T} \alpha_i \tau_i \right\|^2 \leq \left( \sum_{i=1}^{T} \alpha_i \|\tau_i\| \right)^2 \leq C,$$

which means that $\theta^T_{\mathrm{Add}}$ is within the fine-tuning regime and satisfies the local smoothness assumption. Hence, if $x \sim \mathcal{D}_i$

$$\mathcal{L}_i(\theta^T_{\mathrm{Add}}) - \mathcal{L}_i(\theta_i) \leq \left\langle \theta^T_{\mathrm{Add}} - \theta_i, \frac{\partial \mathcal{L}_i(x, \theta_i)}{\partial \theta} \right\rangle + \frac{L_i}{2} \left\| \theta^T_{\mathrm{Add}} - \theta_i \right\|^2 \qquad \text{(Assumption B.3)}$$

$$= \left\langle \sum_{j=1}^{T} \alpha_j \tau_j - \tau_i, \frac{\partial \mathcal{L}_i(x, \theta_i)}{\partial \theta} \right\rangle + \frac{L_i}{2} \left\| \sum_{j=1}^{T} \alpha_j \tau_j - \tau_i \right\|^2 \qquad \text{(Assumption B.2)}$$

To bound the second norm term, we reassign the subscript $\alpha_k := \alpha_i$ as the coefficient for the $i$-th task to avoid confusion with the summation indices. Next we define $t_j := \alpha_j \ (j \neq k), t_k := \alpha_k - 1$. Since $\sum_i \alpha_i = 1$, we have $t_k + \sum_{j \neq k} t_j = 0$. Then,

$$\left\| \sum_{j=1}^{T} \alpha_j \tau_j - \tau_i \right\|^2 = \left\| \sum_{j=1}^{T} \alpha_j \tau_j - \tau_k \right\|^2$$

$$= \left\| \sum_{i=1}^{n} t_i \tau_i \right\|^2$$

$$\leq \sum_{i=1}^{n} t_i^2 \tau_i^2 + 2 \sum_{1 \leq i < j \leq n} |t_i t_j| \cdot |\langle \tau_i, \tau_j \rangle|$$

$$\leq C \left( \sum_{i=1}^{n} t_i^2 + 2\epsilon \sum_{1 \leq i < j \leq n} |t_i t_j| \right) \tag{17}$$

---

[1]This is NOT assuming all task vectors are near-orthogonal. Depending on the benchmark, $\epsilon$ can be as large as 1 where task vectors are completely aligned (or flipped).

By Assumption B.4, $t_k \leq 0$. Besides, we have $-t_k = t_1 + \cdots + t_{k-1} + t_{k+1} + \cdots + t_n = \sum_{i \neq k} t_i$. We can bound the cross-product term as follows:

$$\sum_{1 \leq i < j \leq n} |t_i t_j| = -t_k \sum_{i \neq k} |t_i| + \sum_{1 \leq i < j \leq n, i \neq k, j \neq k} t_i t_j$$

$$= t_k^2 + \sum_{1 \leq i < j \leq n, i \neq k, j \neq k} t_i t_j$$

$$= t_k^2 + \frac{1}{2} \left( \left( \sum_{i \neq k} t_i \right)^2 - \sum_{i \neq k} t_i^2 \right)$$

$$= t_k^2 + \frac{1}{2} \left( t_k^2 - \left( \sum_i t_i^2 - t_k^2 \right) \right)$$

$$= 2t_k^2 - \frac{1}{2} \sum_i t_i^2$$

Plug it back into Eq. (17), we have

$$\sum_i t_i^2 + 2\epsilon \sum_{1 \leq i < j \leq n} |t_i t_j| = \sum_i t_i^2 + 2\epsilon \left( 2t_k^2 - \frac{1}{2} \sum_i t_i^2 \right)$$

$$= (1 - \epsilon) \sum_i t_i^2 + 4\epsilon t_k^2$$

$$= (1 - \epsilon) \left[ \sum_{i \neq k} \alpha_i^2 + (\alpha_k - 1)^2 \right] + 4\epsilon (\alpha_k - 1)^2$$

$$= (1 - \epsilon) \left[ \sum_i \alpha_i^2 - 2\alpha_k + 1 \right] + 4\epsilon (\alpha_k - 1)^2$$

$$\leq (1 - \epsilon) \cdot 2 + 4\epsilon \cdot 1 \qquad \text{(Assumption B.4)}$$

$$= 2 + 2\epsilon$$

To conclude, we have

$$\mathcal{L}_i(\theta_{\text{Add}}^T) - \mathcal{L}_i(\theta_i) \leq \frac{L_i C}{2}(2 + 2\epsilon) = L_i C(1 + \epsilon). \qquad \square$$

Thm. B.5 shows that as long as the task vectors reside in the fine-tuning regime and task vectors are dissimilar enough, then a single model obtained by model merging simultaneously performs comparably well on all the tasks. The local smoothness constant $L_i$ in the generalization bound implies that a flatter minima is preferred in model merging (Iurada et al., 2025; Lee et al., 2025), which also related to the Fisher weighted averaging method (Matena & Raffel, 2022) as $\mathbf{H}$ agrees with the Fisher information matrix when $\ell$ is the cross-entropy loss which is a log-likelihood.

The following corollary shows that the softmax-activated Autoencoder bases formulation will not introduce additional performance gap in the upper bound of vector addition.

**Corollary B.6** (Basis addition reduces to task addition). *Let $B_m = \sum_{j=1}^T \mathbf{W}_e[j, m]\tau_j$ as defined in Eq. (8). Define the basis-merged model $\theta_{\text{Add}}^M = \theta_0 + \sum_{m=1}^M \alpha_m B_m$. For every $i \in [T]$,*

$$\mathcal{L}_i(\theta_{\text{Add}}^M) - \mathcal{L}_i(\theta_i) \leq L_i C (1 + \epsilon). \tag{18}$$

*Therefore, bases addition share the same generalization bound as standard task addition.*

*Proof.* Define effective task weights $\phi_j = \sum_{m=1}^M \alpha_m \mathbf{W}_e[j, m]$. Since $\alpha \in \Delta^{M-1}$ and each $\mathbf{W}_e[:, m] \in \Delta^{T-1}$, we have $\phi \in \Delta^{T-1}$ ($\phi_j \geq 0$ and $\sum_{j=1}^T \phi_j = 1$). Hence

$$\theta_{\text{Add}}^M - \theta_0 = \sum_{m=1}^M \alpha_m \sum_{j=1}^T \mathbf{W}_e[j, m]\tau_j = \sum_{j=1}^T \left( \sum_{m=1}^M \alpha_m \mathbf{W}_e[j, m] \right) \tau_j = \sum_{j=1}^T \phi_j \tau_j,$$

so $\theta_{\mathrm{Add}}^M$ is a convex combination of task vectors.

By Assumption B.2, $\|\tau_j\|^2 \leq C$. Using $\phi \in \Delta^{T-1}$,

$$\|\theta_{\mathrm{Add}}^M - \theta_0\| = \Big\| \sum_j \phi_j \tau_j \Big\| \leq \sum_j \phi_j \|\tau_j\| \leq \sqrt{C},$$

so $\|\theta_{\mathrm{Add}}^M - \theta_0\|^2 \leq C$, placing $\theta_{\mathrm{Add}}^M$ in the local region where Assumption B.3 applies.

The claim follows immediately from Thm. B.5. $\qquad\square$

### B.2.3 OOD Generalization with Task Vectors & Bases

Though similar tasks represented by $\epsilon$ may hurt addition and negation (see Thm. B.9), it is possible to achieve OOD generalization given insights from (Tripuraneni et al., 2020; Hu et al., 2024).

**Theorem B.7** (Out-of-Distribution Generalization). *Given a collection of source task vectors $\mathcal{S} = \{\tau_1, \tau_2, \ldots, \tau_T\}$ and a target task vector with $\|\tau_{\mathrm{tar}}\|^2 \leq C$. If $\exists i \in [T]$ such that $\langle \tau_{\mathrm{tar}}, \tau_i \rangle \geq \gamma C$ for $0 < \gamma \leq 1$, then there exists a merging scheme $\alpha_i, i \in [T]$ such that for the merged model $\theta_{\mathrm{Add}}^T = \theta_0 + \sum_{i=1}^T \alpha_i \tau_i$,*

$$\mathcal{L}_{\mathrm{tar}}(\theta_{\mathrm{Add}}^T) \leq \mathcal{L}_{\mathrm{tar}}(\theta_{\mathrm{tar}}) + L_{\mathrm{tar}} C (1 - \gamma). \tag{19}$$

*Proof.* Let $i^* = \arg\max_{i \in [T]} \langle \tau_{\mathrm{tar}}, \tau_i \rangle$ and choose $\alpha_{i^*} = 1, \alpha_j = 0, \forall j \neq i^*$. Clearly $\langle \tau_{\mathrm{tar}}, \tau_{i^*} \rangle \geq \beta C$ and $\theta_{\mathrm{Add}}^T = \theta_0 + \tau_{i^*}$. It is also easy to check that $\|\theta_{\mathrm{Add}}^T - \theta_{\mathrm{tar}}\| \leq 4C$. So by the local smoothness assumption of $\mathcal{L}_{\mathrm{tar}}$, we have

$$\begin{aligned}
\mathcal{L}_{\mathrm{tar}}(\theta_{\mathrm{Add}}^T) - \mathcal{L}_{\mathrm{tar}}(\theta_{\mathrm{tar}}) &\leq \frac{L_{\mathrm{tar}}}{2} \|\theta_{\mathrm{Add}}^T - \theta_{\mathrm{tar}}\|^2 \\
&= \frac{L_{\mathrm{tar}}}{2} \|\tau_{i^*} - \tau_{\mathrm{tar}}\|^2 \\
&\leq \frac{L_{\mathrm{tar}}}{2} \left( C - 2\langle \tau_{\mathrm{tar}}, \tau_{i^*} \rangle + C \right) \\
&\leq L_{\mathrm{tar}} C (1 - \gamma). \qquad\square
\end{aligned}$$

This implies when $\gamma$, which roughly corresponds to the similarity of the two task vectors, is large enough, the gap between $\mathcal{L}_{\mathrm{tar}}(\theta_{\mathrm{Add}}^T)$ and $\mathcal{L}_{\mathrm{tar}}(\theta_{\mathrm{tar}})$ is small, so we can use the combination of similar task vectors to achieve similar generalization performance for tasks that are OOD w.r.t. the source models.

**Theorem B.8** (OOD generalization via basis atoms). *Assume the target task vector $\tau_{\mathrm{tar}}$ and all source tasks are nonnegatively aligned, i.e., $\langle \tau_{\mathrm{tar}}, \tau_i \rangle \geq 0$ for every $i \in [T]$, and $\exists m^* \in [M]$ such that $\mathbf{W}_e[i^*, m^*] \geq \rho$ for some $\rho \in (0, 1]$ where $i^* = \arg\max_{i \in [T]} \langle \tau_{\mathrm{tar}}, \tau_i \rangle$. Denote $\theta_{\mathrm{Add}}^M = \theta_0 + \sum_{m=1}^M \alpha_m B_m$ with $\alpha \in \Delta^{M-1}$ and $B_m = \sum_{j=1}^T \mathbf{W}_e[j, m] \tau_j$ as defined in Eq. (8), then there exists a choice of $\alpha$ such that*

$$\mathcal{L}_{\mathrm{tar}}(\theta_{\mathrm{Add}}^M) \leq \mathcal{L}_{\mathrm{tar}}(\theta_{\mathrm{tar}}) + L_{\mathrm{tar}} C \left( 1 - \rho \gamma \right). \tag{20}$$

*Proof.* First note that the existential coverage condition for $m^*$ is always satisfied for softmax encoders by picking $\rho = \max_{m \in [M]} \mathbf{W}_e[i^*, m]$.

Now consider the construction with $\alpha_{m^*} = 1$ and $\alpha_m = 0$ for $m \neq m^*$, so that $\theta_{\mathrm{Add}}^M = \theta_0 + B_{m^*}$.

Since $\|B_{m^*}\|^2 = \|\sum_{j=1}^T \mathbf{W}_e[j, m^*] \tau_j\|^2 \leq C$, it is easy to see $\|\theta_{\mathrm{Add}}^M - \theta_{\mathrm{tar}}\| \leq 4C$. So by the local smoothness assumption of $\mathcal{L}_{\mathrm{tar}}$, we have

$$\mathcal{L}_{\mathrm{tar}}(\theta_{\mathrm{Add}}^M) - \mathcal{L}_{\mathrm{tar}}(\theta_{\mathrm{tar}}) \leq \frac{L_{\mathrm{tar}}}{2} \|\theta_{\mathrm{Add}}^M - \theta_{\mathrm{tar}}\|^2 = \frac{L_{\mathrm{tar}}}{2} \|B_{m^*} - \tau_{\mathrm{tar}}\|^2.$$

Expand and bound the quadratic term:

$$\|B_{m^*} - \tau_{\mathrm{tar}}\|^2 = \|B_{m^*}\|^2 + \|\tau_{\mathrm{tar}}\|^2 - 2\langle \tau_{\mathrm{tar}}, B_{m^*} \rangle \leq 2C - 2\langle \tau_{\mathrm{tar}}, B_{m^*} \rangle,$$

using $\|B_{m^*}\| \leq \sqrt{C}$ and $\|\tau_{\text{tar}}\| \leq \sqrt{C}$, now,

$$
\begin{aligned}
\langle \tau_{\text{tar}}, B_{m^*} \rangle &= \sum_{j=1}^{T} \mathbf{W}_e[j, m] \langle \tau_{\text{tar}}, \tau_j \rangle \\
&\geq \mathbf{W}_e[i^*, m^*] \langle \tau_{\text{tar}}, \tau_{i^*} \rangle \qquad \text{(nonnegative source target alignment)} \\
&\geq \rho \gamma C
\end{aligned}
$$

Therefore,

$$
\|B_{m^*} - \tau_{\text{tar}}\|^2 \leq 2C - 2\rho\gamma C = 2C(1 - \rho\gamma),
$$

and hence

$$
\mathcal{L}_{\text{tar}}(\theta_{\text{Add}}^M) - \mathcal{L}_{\text{tar}}(\theta_{\text{tar}}) \leq \frac{L_{\text{tar}}}{2} 2C(1 - \rho\gamma) = L_{\text{tar}} C (1 - \rho\gamma).
$$

$\square$

When the softmax encoder is annealed with smaller $\tau$ and columns of $\mathbf{B}$ converges to one-hot, $\rho \approx 1$ and the bound recovers the original $L_{\text{tar}} C(1 - \beta)$ rate.

### B.2.4 TASK NEGATION & BASIS NEGATION

**Theorem B.9** (Task Negation for Unlearning). *Let $0 < \epsilon \leq 1$ be a universal constant such that $\forall i \neq j, |\cos(\tau_i, \tau_j)| \leq \epsilon$, and $\theta_{\text{Neg,i}} = \theta_0 - \alpha_i \tau_i$ be the model parameter used for unlearning task $i$. Then $\forall j \neq i$,*

$$
\mathcal{L}_j(\theta_{\text{Neg,i}}) - \mathcal{L}_j(\theta_0) \leq L_j C \left( \frac{3}{2} + \epsilon \right). \tag{21}
$$

*Proof.* First, note that

$$\mathcal{L}_j(\theta_{\text{Neg,i}}) - \mathcal{L}_j(\theta_0) \leq \mathcal{L}_j(\theta_{\text{Neg,i}}) - \mathcal{L}_j(\theta_j) + \mathcal{L}_j(\theta_j) - \mathcal{L}_j(\theta_0) \leq \mathcal{L}_j(\theta_{\text{Neg,i}}) - \mathcal{L}_j(\theta_j) + |\mathcal{L}_j(\theta_0) - \mathcal{L}_j(\theta_j)|$$

We will upper bound the last two terms separately. To bound $\mathcal{L}_j(\theta_{\text{Neg,i}}) - \mathcal{L}_j(\theta_j)$, note that $\|\theta_{\text{Neg,i}} - \theta_j\|^2 = \|\alpha_i \tau_i + \tau_j\|^2 \leq (\alpha_i \|\tau_i\| + \|\tau_j\|)^2 \leq 4C$, due to the local smoothness of $\mathcal{L}_j$ around $\theta_j$ and the fact that $\partial \mathcal{L}_j(\theta_j)/\partial\theta = \mathbf{0}$, we have

$$\mathcal{L}_j(\theta_{\text{Neg,i}}) - \mathcal{L}_j(\theta_j) \leq \frac{L_j}{2} \|\theta_{\text{Neg,i}} - \theta_j\|^2 = \frac{L_j}{2} \|\alpha_i \tau_i + \tau_j\|^2 \leq \frac{L_j}{2} \left( \alpha_i^2 C + 2\langle \tau_i, \tau_j \rangle + C \right) \leq L_j C(1 + \epsilon).$$

Similarly, for the second term, we have

$$
|\mathcal{L}_j(\theta_0) - \mathcal{L}_j(\theta_j)| \leq \frac{L_j}{2} \|\theta_0 - \theta_j\|^2 \leq \frac{L_j C}{2}.
$$

Combine both above, leading to

$$
\mathcal{L}_j(\theta_{\text{Neg,i}}) - \mathcal{L}_j(\theta_0) \leq L_j C(1 + \epsilon) + \frac{L_j C}{2} = L_j C \left( \frac{3}{2} + \epsilon \right),
$$

as desired. $\square$

Since $C$ is small due to fine-tuning, $\mathcal{L}_j(\theta_{\text{Neg,i}}) \approx \mathcal{L}_j(\theta_0)$, which means that the negation of a task for forgetting will not adversely impact the performance of other unrelated tasks, which has been shown empirically in (Ilharco et al., 2022), in contrast to other classic unlearning methods like gradient ascent.

**Theorem B.10** (Basis Negation for Unlearning). *Let $\hat{\mathbf{T}}$ be the task vector matrix reconstructed by basis vectors defined by Eq. (10), and write $\widehat{\tau}_i$ for its $i$-th column and per column reconstruction error $e_i := \widehat{\tau}_i - \tau_i$. Define the negation model $\theta_{\text{Neg,i}} := \theta_0 - \alpha_i \widehat{\tau}_i$. Then for every $j \neq i$,*

$$
\mathcal{L}_j(\theta_{\text{Neg,i}}) - \mathcal{L}_j(\theta_0) \leq L_j C \left( \frac{5}{2} + 2\epsilon \right) + L_j \|e_i\|_2^2. \tag{22}
$$

*Proof.* Decompose as in the original proof of Thm. B.9:

$$\mathcal{L}_j(\theta_{\text{Neg},i}) - \mathcal{L}_j(\theta_0) \leq \big(\mathcal{L}_j(\theta_{\text{Neg},i}) - \mathcal{L}_j(\theta_j)\big) + \big|\mathcal{L}_j(\theta_0) - \mathcal{L}_j(\theta_j)\big|.$$

By Assumption B.3 at $\theta_j$ and $\nabla\mathcal{L}_j(\theta_j) = \mathbf{0}$ from Assumption B.2,

$$\mathcal{L}_j(\theta_{\text{Neg},i}) - \mathcal{L}_j(\theta_j) \;\leq\; \frac{L_j}{2}\left\|\theta_{\text{Neg},i} - \theta_j\right\|^2 = \frac{L_j}{2}\left\|\alpha_i(\tau_i + e_i) + \tau_j\right\|^2.$$

Expand the square:

$$\begin{aligned}
\frac{L_j}{2}\left\|\alpha_i(\tau_i + e_i) + \tau_j\right\|^2 &= \frac{L_j}{2}\left\|(\alpha_i\tau_i + \tau_j) + \alpha_i e_i\right\|^2 \\
&\leq \frac{L_j}{2}\left(\|\alpha_i\tau_i + \tau_j\| + \|e_i\|\right)^2 && (\alpha_i \leq 1) \\
&\leq L_j\big(\underbrace{\|\alpha_i\tau_i + \tau_j\|^2}_{(\star)} + \|e_i\|^2\big) && ((a+b)^2 \leq 2(a^2 + b^2))
\end{aligned}$$

The baseline term $(\star)$ is the same as in the original task vector negation proof in Thm. B.9, where it yields $L_j\,(\star) \;\leq\; 2L_j C(1 + \epsilon)$.

Finally, as in Thm. B.9, $\big|\mathcal{L}_j(\theta_0) - \mathcal{L}_j(\theta_j)\big| \;\leq\; \frac{L_j}{2}\|\theta_0 - \theta_j\|^2 \;\leq\; \frac{L_j C}{2}$. Summing the two parts completes the bound. $\square$

Compared to the original negation bound $L_j C(\frac{3}{2} + \epsilon)$, the difference is increased constants for $L_j C$ term another term only depending on the reconstruction error $\|e_i\|$. If the autoencoder (or PCA) bases reconstructs the $i$-th task vector well (small $\|e_i\|$), the penalty is negligible, recovering the original guarantee.

**Corollary B.11** (Optimal Autoencoder Bases Negation). *Let the error matrix be $\mathbf{E} := \hat{\mathbf{T}} - \mathbf{T}$ and suppose the trained autoencoder attains the spectral lower bound in Eq. (7). Then for all $j \neq i$,*

$$\mathcal{L}_j(\theta_{\text{Neg},i}) - \mathcal{L}_j(\theta_0) \;\leq\; L_j C\left(\frac{5}{2} + 2\epsilon\right) \;+\; L_j\lambda_{M+1}(\mathbf{G}). \tag{23}$$

*Proof.* Since the autoencoder attains the spectral lower bound in Eq. (7), we have

$$\|\mathbf{E}\|_2^2 \;=\; \lambda_{M+1}(\mathbf{G}).$$

For the $i$-th column error $e_i = \mathbf{E}\,\mathbf{e}_i$ (with $\mathbf{e}_i$ the $i$-th standard basis vector),

$$\|e_i\|_2^2 \;=\; \|\mathbf{E}\mathbf{e}_i\|_2^2 \;\leq\; \|\mathbf{E}\|_2^2\,\|\mathbf{e}_i\|_2^2 \;=\; \|\mathbf{E}\|_2^2 \;=\; \lambda_{M+1}(\mathbf{G}).$$

Substituting $\|e_i\|_2^2 \leq \lambda_{M+1}(\mathbf{G})$ into the bound from Thm. B.10 yields the desired upper bound as claimed. $\square$

# C  ADDITIONAL EXAMPLES OF ADAPTING TASK VECTOR METHODS TO BASES

**1-to-1 task dataset correspondence.** Both Fisher weighting (Matena & Raffel, 2022) and Reg-Mean (Jin et al., 2022) also implicitly assume a one-to-one mapping between each task dataset $D_i$ and its task vector $\tau_i$ (or equivalently fine tuned model $\theta_i$). For Fisher merging, this is because each task contributes its own Fisher matrix $\mathbf{F}_i$ computed on validation data from $D_i$. For RegMean, each task contributes a validation covariance $\mathbf{X}_i^\top \mathbf{X}_i$, again computed directly from $D_i$. Thus, in the original $T$ tasks setting, every task has its own dataset-level statistics that align exactly with its task vector.

In the basis setting, this direct correspondence is broken: bases are mixtures of tasks rather than stand-alone vectors. To preserve compatibility, we follow the same idea as in our adaptation of Localize-and-Stitch in Sec. 3.2.2. Specifically, we construct basis-level validation statistics by taking encoder-weighted combinations of the original per-task quantities. For Fisher, this yields a basis-level Fisher $\widetilde{\mathbf{F}}_m = \sum_{i=1}^T \mathbf{W}_e[i, m]\,\mathbf{F}_i$ that aggregates information from all tasks according to their encoder weights. For RegMean, the per-task covariance matrices are aggregated into $\widetilde{\mathbf{G}}_m = \sum_{i=1}^T \mathbf{W}_e[i, m]\,\mathbf{X}_i^\top \mathbf{X}_i$. These mixtures define new per-basis statistics, which can then be used in the same closed-form merging rules as the original methods.

Table 5: Comparison of original task-vector inference formulations and their basis-setting counterparts during addition for multitask learning.

| Method | $T$ tasks setting | $M$ bases setting |
|---|---|---|
| Fisher merge | $\theta_{\text{Add}} = \dfrac{\sum_{i=1}^{T} \mathbf{F}_i \theta_i}{\sum_{i=1}^{T} \mathbf{F}_i}$ | $\theta_{\text{Add}} = \dfrac{\sum_{m=1}^{M} \widetilde{\mathbf{F}}_m (\theta_0 + B_m)}{\sum_{m=1}^{M} \widetilde{\mathbf{F}}_m}, \ \widetilde{\mathbf{F}}_m = \sum_{i=1}^{T} \mathbf{W}_e[i,m] \mathbf{F}_i$ |
| RegMean | $\theta_{\text{Add}} = \Big( \sum_{i=1}^{T} \mathbf{X}_i^\top \mathbf{X}_i \Big)^{-1} \Big( \sum_{i=1}^{T} \mathbf{X}_i^\top \mathbf{X}_i \theta_i \Big)$ | $\theta_{\text{Add}} = \Big( \sum_{m=1}^{M} \widetilde{\mathbf{G}}_m \Big)^{-1} \Big( \sum_{m=1}^{M} \widetilde{\mathbf{G}}_m (\theta_0 + B_m) \Big), \ \widetilde{\mathbf{G}}_m = \sum_{i=1}^{T} \mathbf{W}_e[i,m] \mathbf{X}_i^\top \mathbf{X}_i$ |
| Localize & Stitch | $S_i = \arg\min_{S \in \mathbb{R}^d} \mathcal{L}_i(\theta_0 + \sigma(S) \odot \tau_i) + \lambda \|\sigma(S)\|_1$ | $S_m = \arg\min_{S \in \mathbb{R}^d} \Big( \sum_{i=1}^{T} \mathbf{W}_e[i,m] \mathcal{L}_i(\theta_0 + \sigma(S) \odot B_m) \Big) + \lambda \|\sigma(S)\|_1$ |
| Task Arithmetic | $\theta_{\text{Add}} = \theta_0 + \sum_{i=1}^{T} \alpha_i \tau_i$ | $\theta_{\text{Add}} = \theta_0 + \sum_{m=1}^{M} \alpha_m B_m$ |
| TIES | $\theta_{\text{Add}} = \theta_0 + \alpha \text{TrimElectMerge}(\tau_1, \ldots, \tau_T)$ | $\theta_{\text{Add}} = \theta_0 + \alpha \text{TrimElectMerge}(B_1, \ldots, B_m)$ |
| AdaMerging | $\min_{\lambda_1, \ldots, \lambda_T} \sum_{x_i \in D_t} H \Big[ f \Big( x_i; \theta_0 + \big\{ \sum_{t=1}^{T} \lambda_t^l \tau_t^l \big\}_{l=1}^{L} \Big) \Big]$ | $\min_{\lambda_1, \ldots, \lambda_M} \sum_{x_i \in \widetilde{D}_t} H \Big[ f \Big( x_i; \theta_0 + \big\{ \sum_{m=1}^{M} \lambda_m^l B_m^l \big\}_{l=1}^{L} \Big) \Big]$ |
| aTLAS | $\min_{\Lambda_1, \ldots, \Lambda_T} \sum_{(x_i, y_i) \in D_t} \Big[ \ell(f(x_i; \theta_0 + \sum_{i=1}^{T} \Lambda_i \tau_i), y_i) \Big]$ | $\min_{\Lambda_1, \ldots, \Lambda_M} \sum_{(x_i, y_i) \in \widetilde{D}_t} \Big[ \ell(f(x_i; \theta_0 + \sum_{m=1}^{M} \Lambda_m B_m), y_i) \Big]$ |

**Others.** For other methods, adapting task-vector methods to a basis representation does not change their mathematical form: Task Arithmetic (Ilharco et al., 2022) and TIES (Yadav et al., 2024) still involve coefficient searching over nonnegative coefficients using validation data, while AdaMerging (Yang et al., 2023) and aTLAS (Zhang et al., 2024) continue to solve unconstrained coefficient-learning problems either on a small unlabeled or labeled dataset. The only modifications are (i) replacing task vectors $\tau_i$ with bases $B_m$, and (ii) replacing $D_t$ with subsampled datasets $\widetilde{D}_t$.

# D EMPIRICAL VERIFICATION OF THEORETICAL RESULTS

We primarily focus on verifying task addition Thm. B.5 where $M = 100\%$ and examine the relationship between the loss gap and key constants throughout our theoretical statements in Sec. B.2.

## D.1 INSPECTION OF KEY CONSTANTS

Table 6: Ratio and Norm Task Vector for Different Models and Datasets

| Model | Dataset | Ratio% | Norm Task Vector |
|---|---|---|---|
| laion2b_e16 | MNIST | 0.46 | 2.18 |
| | EuroSAT | 0.45 | 2.16 |
| | Cars | 0.54 | 2.52 |
| | DTD | 0.39 | 1.81 |
| | GTSRB | 0.49 | 2.32 |
| | RESISC45 | 0.54 | 2.55 |
| | SUN397 | 0.65 | 3.03 |
| | SVHN | 0.56 | 2.64 |
| laion2b_s34b_b79k | MNIST | 0.42 | 2.30 |
| | EuroSAT | 0.42 | 2.27 |
| | Cars | 0.48 | 2.59 |
| | DTD | 0.33 | 1.79 |
| | GTSRB | 0.45 | 2.44 |
| | RESISC45 | 0.48 | 2.63 |
| | SUN397 | 0.58 | 3.18 |
| | SVHN | 0.51 | 2.76 |

**Task Vector Norm** $C$ Two openclip checkpoints are details can be found from `https://github.com/mlfoundations/open_clip/blob/main/docs/PRETRAINED.md` where laion2b_s34b_b79k is reported to be trained with larger batch size and learning rate, while two models share the same training data LAION-2B (Schuhmann et al., 2022). In Tab. 6, we

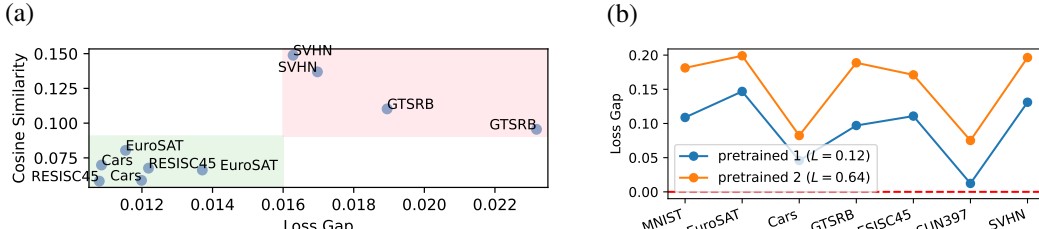

Figure 5: (a) Task vector similarity vs. $\mathcal{L}_{\text{MNIST}}(\theta_{\text{Add}}^2) - \mathcal{L}_{\text{MNIST}}(\theta_{\text{MNIST}})$, where $\theta_{\text{Add}}^2 = \theta_0 + 0.5\tau_{\text{MNIST}} + 0.5\tau_{\text{task}}$. This figure includes two different set of CLIP ViT/B-32 task vectors. The pink shade includes the high similarity high loss gap region, and the green shade is the low similarity low loss gap region. This implies larger task similarity $\epsilon$ is harmful for addition. (b) $\mathcal{L}_{\text{DTD}}(\theta_{\text{Add}}^2) - \mathcal{L}_{\text{DTD}}(\theta_{\text{DTD}})$ by merging $\tau_{\text{DTD}}$ with other task vectors, setting scaling coefficient as $0.5$. Two colored pretrained checkpoints have different local smoothness values.

reported the task vector norm and the ratio of task vector norm over the pretrained model norm, which is very small across datasets and models.

We elaborate on the connection of small task vector norm $C$ requirement with previous literature. (Ilharco et al., 2022) in its Figure 7 demonstrates that the performance of merging task vectors derived from intermediate checkpoints, far before model convergence, is close to the performance of merging converged task vectors. These intermediate checkpoints typically have smaller norms due to fewer optimization steps, so a small $C$ appears sufficient for the success of task addition. On the other hand, Figure 6 of (Ilharco et al., 2022) also indicated that a smaller learning rate is more important for task addition than for standard single-task fine-tuning, which implies that a smaller $C$ is also necessary.

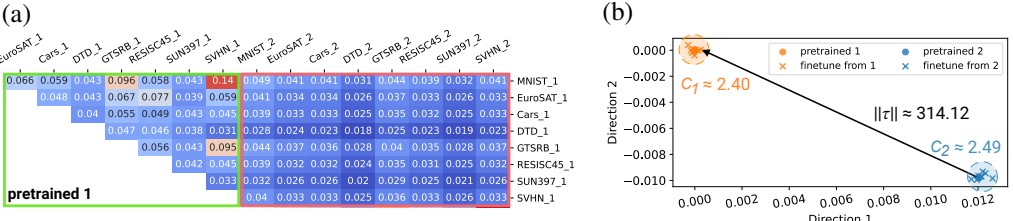

Figure 6: (a) Task vector similarity matrix for two checkpoints. The left green box represents the task similarity for vectors all derived from fine-tuning pretrained model 1. The right pink box represents the similarity values for task vectors from two different checkpoints, which corresponds to small $\epsilon$ in the 50/50 row of Tab. 7. (b) Task vector norms in different settings. Since the distance of two pretrained models are much larger than the distance between the pretrained model and their own fine tuned model , in Tab. 7 if we subtract pretrained 2 from any finetune 1, $\|\tau\|$'s upper bound $C$ is huge, leading to the merging failure. For visualization purpose, we show two randomly selected dimensions, but the numbers for $C_1, C_2, \|\tau\|$ are directly computed from high-dimensional vectors.

**Task Vector Similarity** $\epsilon$ To verify how task vector similarity $\epsilon$ impacts the performance, we conduct the experiment shown in Fig. 5a. We merge the MNIST task vector with each of the other task vectors, all having similar norms ranging from $[2, 3]$ (see Tab. 6), and set the scaling coefficient $\alpha$ to be $0.5$. In this setting, we approximately control all constants in Thm. B.5, including $L$, $\alpha$, and $C$, and observe that highly similar tasks, such as digit classification in MNIST, SVHN, and GTSRB, lead to larger loss gaps or worse performance for MNIST compared to less related tasks.

**Interaction between $C$ and $\epsilon$** We provide additional evidence that both $C$ and $\epsilon$ must be constrained for the success of task vector addition. In Tab. 7, we collected task vectors for 8 tasks from two CLIP checkpoints pretrained with different hyperparameters. From this table, we observe that successful task addition reveals the identity of the task vectors. For optimal merging performance, we should only add task vectors fine-tuned from the same checkpoint, as any mixture of task vectors from different checkpoints will cause a significant performance drop. The above empirical observation consolidates our Assumption B.2 that task vectors should reside in the same fine-tuning regime.

Table 7: Task vector mixing performance, which is the average of all task test accuracies evaluated with the merged model. Numbers 1 and 2 refer to the identities of the pretrained checkpoints. "50 / 50" represents the experiment where 50% of the own task vector is mixed with 50% of task vectors derived from the other pretrained model.

| $\theta_i \setminus \theta_0$ | pretrained 1 | pretrained 2 |
|---|---|---|
| finetune from 1 | **70.83** | 51.71 |
| 50 / 50 | 59.92 | 61.71 |
| finetune from 2 | 54.21 | **71.09** |

To elaborate, from Fig. 6a, although all $\epsilon$ values in the pink box are very low, task addition still fails due to the large $C$ value. From Fig. 6b, we see that with different hyperparameters, the two pretrained models are situated in two local convex basins, and the distance between the two checkpoints is much larger than the task vectors (Tab. 6). Thus, if we create task vectors by subtracting the wrong pretrained checkpoint, the large $C$ value leads to the failure of task addition.

**Local Smoothness $L$**  The local smoothness $L$ is specific to each pretrained model due to differences in their optimization trajectories. Since, as shown in Fig. 6, the differences in $C$ and $\epsilon$ (not $\theta_i$) between two pretrained models are small. In Fig. 5b, we merge the DTD task vector with each of the other task vectors and compare the loss gap between two checkpoints. Because it is not feasible to load $\mathbf{H}(\theta_i) \in \mathbb{R}^{d \times d}$ directly onto the GPU, we estimate $L$ using the power iteration method (Mises & Pollaczek-Geiringer, 1929) to reduce the largest eigenvalue problem to a Hessian-vector product computation. As seen in Fig. 5b, larger local smoothness consistently leads to a larger gap from the optimal loss term across datasets, resulting in worse merging performance.

D.2   EMPIRICAL LOSS GAP IN TASK ADDITION

We report the empirical loss gap using Task Arithmetic as the merging method on several downstream datasets using the ViT-B/16 backbone for CLIP. In Tab. 8, the loss gap is computed as the empirical difference between the loss of the merged model $\mathcal{L}_i(\theta_{\mathrm{Add}})$ and the fine-tuned loss $\mathcal{L}_i(\theta_i)$ as in the left hand side of theorems in Sec. B.2.2.

Table 8: Empirical loss gap between the fine-tuned model and the TA-merged model for each task on ViT-B/16.

| Task$_i$ | $\mathcal{L}_i(\theta_{\mathrm{Add}}) - \mathcal{L}_i(\theta_i)$ |
|---|---|
| MNIST | 0.12 |
| EuroSAT | 1.00 |
| Cars | 0.15 |
| DTD | 1.11 |
| GTSRB | 0.80 |
| RESISC45 | 0.84 |
| SUN397 | 5.02 |
| SVHN | 0.60 |

As shown in Table 8, the loss gap remains small for most tasks except SUN397, which reaches a gap of $5.02$. This demonstrates that although task arithmetic (Ilharco et al., 2022) is a widely used model merging baseline, it can suffer significant performance degradation on certain tasks. Importantly, these empirical loss gaps are *not* computed from the constants appearing in the right-hand side of our theoretical upper bound. When substituting reasonable values, such as $L = 1$, $C = 4$, and $\epsilon = 0.2$, the upper bound becomes $LC(1 + \epsilon) \approx 5$, which approximates the largest observed gap. While this value may seem conservative for common losses like cross-entropy, it is necessary to account for the worst-case scenarios observed in practice. This further supports the relevance of our theoretical framework in explaining and bounding the limitations of task arithmetic in heterogeneous settings.

## E EXPERIMENTAL DETAILS ABOUT BASES CONSTRUCTION

We construct task vector bases using a lightweight AE trained over the collection of task vectors with Adam optimizer (Kingma, 2014). All experiments are conducted on NVIDIA RTX A6000 GPUs with 48GB memory. Unless otherwise specified, we adopt the following default configuration for the AE:

$$M = 4, \text{ steps} = 4000, \text{ lr} = 0.01, \tau = 5.0, \text{ weight decay} = 10^{-6}.$$

### E.1 SENSITIVITY OF TEMPERATURE $\tau$

A key hyperparameter in the encoder is $\tau$, which controls the mixing effect of task vectors when constructing basis vectors with AE.

Fig. 7 shows the encoder weight distributions across different values of the temperature parameter $\tau$. As $\tau$ decreases, the encoder weights become more selective, pushing the representation closer to a one-hot distribution. At $\tau = 5$, the weights are relatively diffuse, with several tasks contributing simultaneously to each basis. By contrast, at $\tau = 1$ and below, the encoder begins to isolate dominant contributors, and at $\tau = 0.8$ the assignments are nearly one-hot. At this low temperature the learned bases also become interpretable. For example, when $\tau = (500, 0.8)$, basis 1 captures GT-SRB, MNIST, and SVHN, which can be interpreted as a digit classification group. Basis 1 and 2 are dominated by Cars and SUN397, which are classification problems very different from other task vectors in the pool. Basis 3 combines RESISC45, DTD, and EuroSAT, corresponding to texture and satellite imagery that are naturally linked through landscape and surface patterns. This progression illustrates that the softmax activation indeed yields semantically meaningful groupings of tasks.

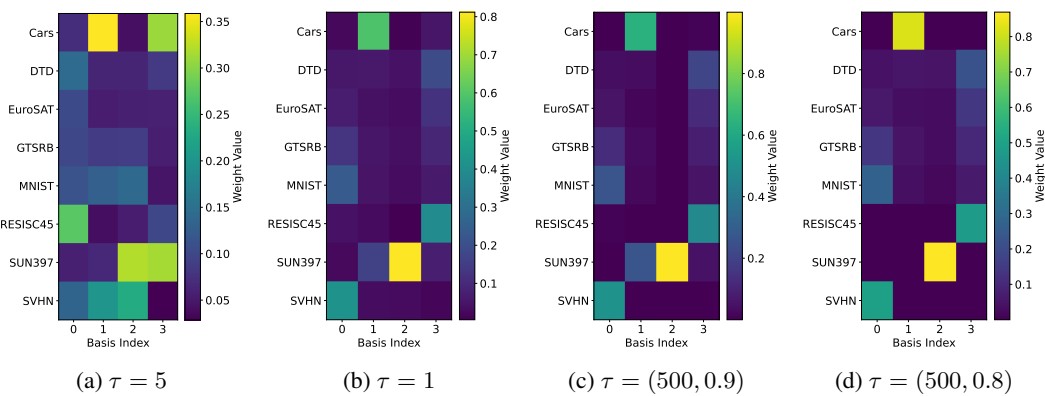

| (a) $\tau = 5$ | (b) $\tau = 1$ | (c) $\tau = (500, 0.9)$ | (d) $\tau = (500, 0.8)$ |

Figure 7: Encoder weight $\mathbf{W}_e$ distributions across different values of the temperature parameter $\tau$ for ViT-B/32 8 tasks setting when $M = 4$. The notation $(500, \eta)$ refers to the annealing $\tau$ setting where every 500 steps the temperature $\tau$ is multiplied by a factor of $\eta$.

The addition performance results for each type of $\tau$ setup are shown in Tab. 9. On one hand, different $\tau$ choices can substantially affect downstream addition performance. In particular, although all settings drive the reconstruction loss close to the theoretical lower bound (note that lower values of $\tau$ require more steps to converge), the actual addition accuracies differ: for example, L&S accuracy vary by as much as 0.09 depending on the schedule. The performance does not grow monotonically with $\tau$ itself, and can vary across setups; for instance, the observed gap to the lower bound depends on the number of training steps, which in turn influences the final accuracy. Therefore, careful hyperparameter tuning of $\tau$ is still required to achieve the best performance. But interestingly, we find that once tuned for one method on a fixed set of task vectors, the same $\tau$ schedule also transfers well across other evaluation metrics (TA results closely track L&S results). On the other hand, negation forgetting metrics are robust to different $\tau$ setups.

### E.2 RUNTIME AND MEMORY ANALYSIS

We first compare the runtime and memory complexity of AE training with PCA for basis construction. Both methods require access to the full task matrix $\mathbf{T} \in \mathbb{R}^{d \times T}$, where $d$ is the parameter

Table 9: Normalized offline addition accuracy and target forgetting accuracy of $\tau$ hyperparameter for ViT-B/32 on 8 tasks with $M = 4$ using AE bases. The spectral lower bound predicted by Eq. (7) is 0.336935.

| $\tau$ | Steps | TA$^+$ | L&S$^+$ | TA$^-$ | Loss |
|---|---|---|---|---|---|
| $(500, 0.8)$ | 30000 | 0.734 | 0.649 | 0.308 | 0.336935 |
| $(500, 0.9)$ | 4000 | 0.722 | 0.640 | 0.306 | 0.338606 |
| 1 | 4000 | 0.736 | 0.707 | 0.308 | 0.336935 |
| 5 | 4000 | 0.744 | 0.733 | 0.307 | 0.336937 |

dimension and $T$ is the number of task vectors. Since typically $d \gg T$, the dominant storage cost for both methods is $O(dT)$. For time complexity, AE first requires one-time computation of the Gram matrix, which costs $O(dT^2)$. Subsequent training then involves $O(T^2)$ operations per optimization step. PCA basis construction is obtained through SVD of the $d \times T$ task matrix, with complexity $O(dT^2)$ too. Thus, in the worst-case analysis both AE and PCA have the same order of time and space complexity. The practical advantage of AE is that, as discussed in Lem. 3.2, after Gram computation we can remove the dependence of $d$ during optimization, making potentially parallel gradient-based optimization feasible on modern GPUs.

We next empirically measure AE training time for different numbers of basis vectors $M$ fixing other hyperparameters. Results are summarized in Tab. 10. For addition experiments, the runtime decreases modestly with larger $M$, which we attribute to more efficient GPU utilization when matrix multiplications involve wider hidden dimensions. This effect is implementation-dependent but does not change the main conclusion that basis processing is almost negligible compared to merging time (see Fig. 2c). For negation, we can see that once the bases are constructed the reconstruction time is also minimal compared to the negation tuning time where we use the grid search over $[0, 0.05, 0.10, \ldots, 1.0]$ (21 grid points) for forgetting experiments.

Table 10: AE steps wall clock time (seconds) for different numbers of basis vectors $M$ with 4000 gradient steps averaged across three runs on ViT-B/32 for 8 tasks. Basis Construction refers to the runtime of solving Eq. (5) and Task Vector Reconstruction refers to the runtime of Eq. (10) given saved bases.

| # Basis | Basis Construction | Addition | Task Vector Reconstruction | Negation |
|---|---|---|---|---|
| $M = 2$ | 28.96 | 10894.54 | 1.11 | 21187.95 |
| $M = 4$ | 22.72 | 11059.96 | 2.12 | 21380.52 |
| $M = 6$ | 17.09 | 11926.79 | 2.61 | 22792.77 |

# F EXPERIMENTAL DETAILS ABOUT BASES ARITHMETIC

## F.1 CHOICE OF SUBSAMPLING STRATEGY AND LEARNING WEIGHTS

Table 11: Effect of subsampling on TA addition accuracy for ViT-B/32 with $M = 50\%$ across different $\tau$ values (8 tasks). We report absolute average accuracy with and without subsampling.

| $\tau$ | $D_i$ | $\widetilde{D_i}$ |
|---|---|---|
| $(500, 0.8)$ | 0.682 | 0.684 |
| $(500, 0.9)$ | 0.667 | 0.667 |
| 1 | 0.682 | 0.681 |
| 5 | 0.689 | 0.689 |

**Subsampling.** During subsampling, we select $n_i \cdot M/T$ examples per original dataset $D_i$ (subsampling stratified by dataset), creating in total $nM$ effective samples in $\cup_{i=1}^{T} \widetilde{D_i}$ compared to $nT$ effective samples in $\cup_{i=1}^{T} D_i$ assuming size of $T$ datasets are the same. In few-shot scenarios, such

as the language experiments (Tab. 2) or OOD experiments (Tab. 3), we similarly adjust the dataset size to $k \cdot M/T$, thereby simulating the stratified subsampling effect. Tab. 11 compares addition results with and without subsampling under TA for ViT-B/32 with $M = 4$ and different $\tau$ schedules. The results show that subsampling has only a minimal impact on accuracy, indicating that tuning on smaller, stratified subsets is sufficient to capture the relevant task relationships while reducing the overall computational cost.

Table 12: Effect of different weighting strategies for AE bases (ViT-B/32, 8 tasks, L&S, $M = 50\%$). We report absolute (normalized) accuracy and average L&S mask sparsity.

| Weighting Strategy | Abs. (Norm.) Acc | Avg. Sparsity |
|---|---|---|
| Uniform | 0.674 (0.727) | 0.913 |
| Fixed (random) | 0.615 (0.669) | **0.944** |
| Encoder Weighted | **0.691 (0.732)** | 0.910 |

**Learning Weights.** Table 12 compares different strategies for setting the basis weights when constructing AE representations. In the uniform case, each task contributes equally ($\mathbf{W}_e[i, m] = 1/T$), while in the fixed random case, each of $M$ basis randomly selects one task to receive weight 1 (others 0). Finally, the encoder-weighted approach uses the learned encoder outputs $\mathbf{W}_e[:, m]$ to define the convex combination of tasks for each basis as in our proposed weighting strategy. We observe that encoder weighting yields the best accuracy under L&S, where accurate localization requires each basis to correspond to a meaningful mixture of tasks. This mapping allows the model to identify parameter regions that explain the combined tasks' performance more effectively than uniform or random weighting. The tradeoff is a slight reduction in sparsity, which modestly increases memory/storage overhead when using CSR format. Nonetheless, the gain in performance strongly supports the use of encoder weights in practice.

### F.2  OFFLINE BASES ADDITION

**Datasets and Metrics.** Define normalized accuracy as absolute accuracy normalized by single task fine tuned performance. For vision experiments, we report the classification accuracy on ViT models (Dosovitskiy, 2020) on Cars (Krause et al., 2013), DTD (Cimpoi et al., 2014), EuroSAT (Helber et al., 2019), GTSRB (Stallkamp et al., 2012), MNIST (LeCun et al., 1998), RESISC45 (Cheng et al., 2017), SVHN (Netzer et al., 2011), and SUN397 (Xiao et al., 2010) for the 8-task setting. For 14 tasks, we additionally include CIFAR100 (Krizhevsky et al., 2009), STL10 (Coates et al., 2011), Flowers102 (Nilsback & Zisserman, 2008), OxfordIIITPet (Parkhi et al., 2012), PCAM (Veeling et al., 2018), FER2013 (Goodfellow et al., 2013), and for 20 tasks, we further include EMNIST (Cohen et al., 2017), CIFAR10 (Krizhevsky et al., 2009), Food101 (Bossard et al., 2014), FashionMNIST (Xiao et al., 2017), RenderedSST2 (Socher et al., 2013) and KMNIST (Clanuwat et al., 2018). For language experiments, we use a 12-task benchmark Gao et al. (2020) with SST2 (Socher et al., 2013), CR (Hu & Liu, 2004), MR (Pang & Lee, 2004), MPQA (Wiebe et al., 2005), TREC (Li & Roth, 2002), SUBJ (Pang & Lee, 2004), QNLI (Wang, 2018), SNLI (Bowman et al., 2015), MNLI (Williams et al., 2017), RTE (Wang, 2018), MRPC (Dolan & Brockett, 2005) and QQP (Sharma et al., 2019) trained on RoBERTa (Liu, 2019) models. We report the F1 metric for MRPC.

**Hyperparameters of Oracle Merging Methods.** For vision experiments, both TA and TIES use validation data for hyperparameter tuning of the isotropic scaling coefficient $\alpha$, with a grid search over $[0, 0.05, 0.10, \ldots, 1.0]$ (21 grid points). For TIES, we additionally set the top-$k$ threshold to 20% and use sum as the merging rule. For L&S, we use the following settings: sigmoid bias = 5, batch size = 16 for ViT-L/14 and 64 otherwise, $\ell_1$ strength = 1, learning rate = $10^{-7}$, 10 training epochs, and sparsity = $10^{-5}$. For language experiments, we perform experiments on 64-shot datasets. TA and TIES adopt the same setup as in vision. For L&S, we use: sigmoid bias = 3, learning rate = $10^{-7}$, $\ell_1$ strength = 0, 10 training epochs, sparsity = $10^{-5}$, and batch size = 8.

**Supplementary Results.** The normalized accuracy results in Tabs. 13 and 14 confirm that our main observations in Sec. 4.1.1 are not simply an artifact of single-task fine-tuning performance.

Even after normalization, AE achieves the best overall results in general, with a clear ordering of AE > RandSelect > PCA across both vision and language tasks. For the TIES method, RandSelect becomes slightly more competitive, and in a few large-task settings it reaches performance on par with AE. The per-dataset results in Figs. 9 to 12 illustrate that AE exhibits a more pronounced advantage in terms of absolute accuracy under L&S, the strongest merging method we evaluate, and its relative advantage grows mildly as the number of tasks increases.

To further strengthen the PCA baseline, we experimented with a simple reweighting scheme in Tab. 13 that converts the task coefficients/PCA loadings $\mathbf{C} = \mathbf{V}_M^\top$ into convex combinations of tasks, denoted by PCA$^{\geq 0}$. For each component $m \in [M]$, let $v_m \in \mathbb{R}^T$ denote the corresponding column of $\mathbf{V}_M$. We define task weights by applying a softmax only to the strictly positive entries of $v_m$:

$$\mathbf{W}_e[i,m] = \begin{cases} \dfrac{\exp(v_{m,i})}{\sum_{j:v_{m,j}>0} \exp(v_{m,j})}, & v_{m,i} > 0, \\ 0, & \text{otherwise,} \end{cases}$$

The resulting per-basis weights $\mathbf{W}_e[:,m] \in \Delta^{T-1}$ ensure that each PCA basis vector can be interpreted as a convex combination of the positively aligned task vectors by completely discarding the negatively aligned ones, thus can be used as the loss weight while training binary masks for L&S. While this positive-softmax variant addresses the concern that our default option uniform PCA ignores the relative contribution of tasks, it still performs significantly worse than AE in practice.

Table 13: Comparison of normalized addition accuracy across ViT models under 8, 14, and 20 vision task settings (Wang et al., 2024a) with bases number $M = 50\%$ of the total tasks.

| Method | ViT-B/16 | | | ViT-B/32 | | | ViT-L/14 | | |
|---|---|---|---|---|---|---|---|---|---|
| | 8 task | 14 task | 20 task | 8 task | 14 task | 20 task | 8 task | 14 task | 20 task |
| TA (Ilharco et al., 2022) | 0.796 | 0.759 | 0.708 | 0.766 | 0.721 | 0.668 | 0.887 | 0.840 | 0.781 |
| RandSelect | 0.695 | 0.706 | 0.672 | 0.698 | 0.703 | 0.674 | 0.808 | 0.811 | 0.763 |
| PCA | 0.538 | 0.629 | 0.622 | 0.578 | 0.633 | 0.645 | 0.686 | 0.729 | 0.705 |
| AE (Ours) | **0.717** | **0.729** | **0.687** | **0.744** | **0.725** | **0.676** | **0.847** | **0.840** | **0.783** |
| TIES (Yadav et al., 2024) | 0.843 | 0.787 | 0.733 | 0.81 | 0.748 | 0.699 | 0.907 | 0.841 | 0.798 |
| RandSelect | 0.719 | 0.718 | 0.672 | 0.712 | **0.717** | **0.689** | 0.847 | 0.821 | **0.779** |
| PCA | 0.540 | 0.629 | 0.622 | 0.576 | 0.633 | 0.654 | 0.686 | 0.728 | 0.707 |
| AE (Ours) | **0.724** | **0.728** | **0.687** | **0.741** | **0.717** | 0.670 | **0.854** | **0.841** | 0.778 |
| L&S (He et al., 2024) | 0.794 | 0.721 | 0.641 | 0.811 | 0.698 | 0.642 | 0.809 | 0.796 | 0.748 |
| RandSelect | 0.575 | 0.569 | 0.465 | 0.580 | 0.527 | 0.467 | 0.696 | 0.722 | 0.643 |
| PCA | 0.540 | 0.482 | 0.440 | 0.489 | 0.438 | 0.389 | 0.692 | 0.629 | 0.581 |
| PCA$^{\geq 0}$ | 0.541 | 0.455 | 0.397 | 0.489 | 0.447 | 0.393 | 0.692 | 0.582 | 0.579 |
| AE (Ours) | **0.694** | **0.716** | **0.687** | **0.732** | **0.718** | **0.679** | **0.767** | **0.780** | **0.780** |

Table 14: Comparison of normalized addition accuracy with RoBERTa-base model on 12 language task benchmark with bases number $M = 25\%$ of the total tasks.

| **TA** (Ilharco et al., 2022) | | | | **TIES** (Yadav et al., 2024) | | | | **L&S** (He et al., 2024) | | | |
|---|---|---|---|---|---|---|---|---|---|---|---|
| 100% | RandSelect | PCA | AE | 100% | RandSelect | PCA | AE | 100% | RandSelect | PCA | AE |
| 0.777 | 0.570 | 0.565 | **0.592** | 0.744 | 0.568 | **0.587** | **0.587** | 0.936 | 0.769 | 0.763 | **0.901** |

**Performance Scaling with $M$.** Fig. 8 plots the offline addition accuracy as the number of basis vectors M increases. We observe that the AE achieves consistently strong performance across all values of $M$, starting higher than both PCA and RandSelect and maintaining steady improvements as $M$ grows. In particular, AE dominates PCA throughout the entire range, with a margin of more than 0.1 absolute accuracy at most values of $M$. RandSelect starts much lower but improves rapidly with larger $M$, eventually approaching and in some cases slightly surpassing AE when $M \geq 6$. This reflects that random bases can capture task diversity when many are available, but their effectiveness is more volatile and requires larger basis sizes to be competitive. In contrast, AE provides

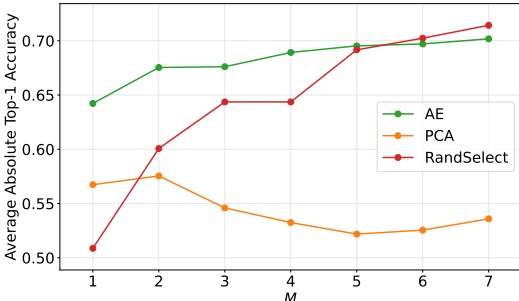

Figure 8: Offline addition accuracy as a function of the number of basis vectors $M$ for different bases construction methods with ViT-B/32 on 8 tasks benchmark.

reliable accuracy gains even with small $M$, making it far more practical in settings where storage or computational budget limits the number of bases.

Table 15: Normalized accuracy across datasets for Llama-3.2-3B (Grattafiori et al., 2024) for 5 task vectors and various compression methods at $M = 40\%$. Best compression results of Task Arithmetic (TA) are bolded.

| Category | Dataset | SFT | TA | AE | PCA | RandSelect |
|---|---|---|---|---|---|---|
| Instruction Following | IFEval (Zhou et al., 2023) | 37.52 | 25.32 | 14.79 | 13.68 | 21.44 |
| Math | GSM8k (Cobbe et al., 2021) | 72.55 | 45.34 | 53.68 | 18.50 | 46.10 |
| | MATH (Hendrycks et al., 2021) | 33.04 | 10.14 | 22.50 | 0.88 | 16.80 |
| | M_MMLU (Lai et al., 2023) | 44.89 | 45.01 | 45.32 | 44.77 | 45.20 |
| | | 43.61 | 44.54 | 48.04 | 46.27 | 47.80 |
| | | 46.45 | 46.11 | 46.07 | 44.99 | 45.92 |
| | | 41.80 | 41.57 | 42.56 | 42.20 | 45.92 |
| Multilingual (fr, de, es, ru) | M_ARC (Lai et al., 2023) | 40.89 | 40.46 | 29.94 | 31.48 | 31.14 |
| | | 38.32 | 36.70 | 37.69 | 36.32 | 38.71 |
| | | 40.09 | 41.54 | 35.07 | 33.70 | 35.41 |
| | | 36.95 | 37.55 | 34.22 | 32.42 | 34.04 |
| | M_Hellaswag (Lai et al., 2023) | 58.67 | 59.56 | 42.09 | 42.40 | 42.71 |
| | | 54.46 | 55.22 | 46.35 | 46.16 | 46.53 |
| | | 60.07 | 61.21 | 43.89 | 43.80 | 44.46 |
| | | 52.63 | 53.89 | 40.76 | 41.03 | 41.22 |
| Coding | Humaneval+ (Chen et al., 2021) | 41.83 | 36.71 | 28.96 | 16.46 | 28.84 |
| | MBPP+ (Austin et al., 2021) | 46.59 | 45.50 | 41.40 | 38.23 | 41.48 |
| Safety | WildGuardTest (Han et al., 2024) | 85.71 | 51.94 | 29.91 | 25.11 | 31.32 |
| | HarmBench (Mazeika et al., 2024) | 89.38 | 39.69 | 27.82 | 24.07 | 30.62 |
| | DoAnythingNow (Shen et al., 2024) | 90.67 | 32.67 | 31.61 | 27.67 | 23.67 |
| | XSTest (Röttger et al., 2023) | 37.56 | 60.22 | 46.89 | 42.22 | 47.11 |
| **Average Normalized Accuracy** | | 100.00 | 72.88 | **63.49** | 47.07 | 63.38 |

**Results on Generative Large Language Models.** In Tab. 15, we use the LLM model merging benchmark (He et al., 2025), evaluating 5 input task vectors per domain on 21 downstream tasks, providing a comprehensive test of LLM abilities with generative tasks. Using Llama-3.2-3B, we tested three basis compression algorithms with $M = 2$. Following He et al. (2025), we report normalized accuracy, defined as the absolute sum of a method's per-category average divided by the absolute sum of the supervised finetuning (SFT) per-category average. We also adopt the recommended scaling coefficient of 0.4 for all addition experiments. Across all settings, AE achieves the

best overall performance, RandSelect ranks second, and PCA is consistently the worst. Notably, AE preserves 87% of the full task vector performance while using only 40% of the compute and storage, which is particularly important for scaling to larger foundation models.

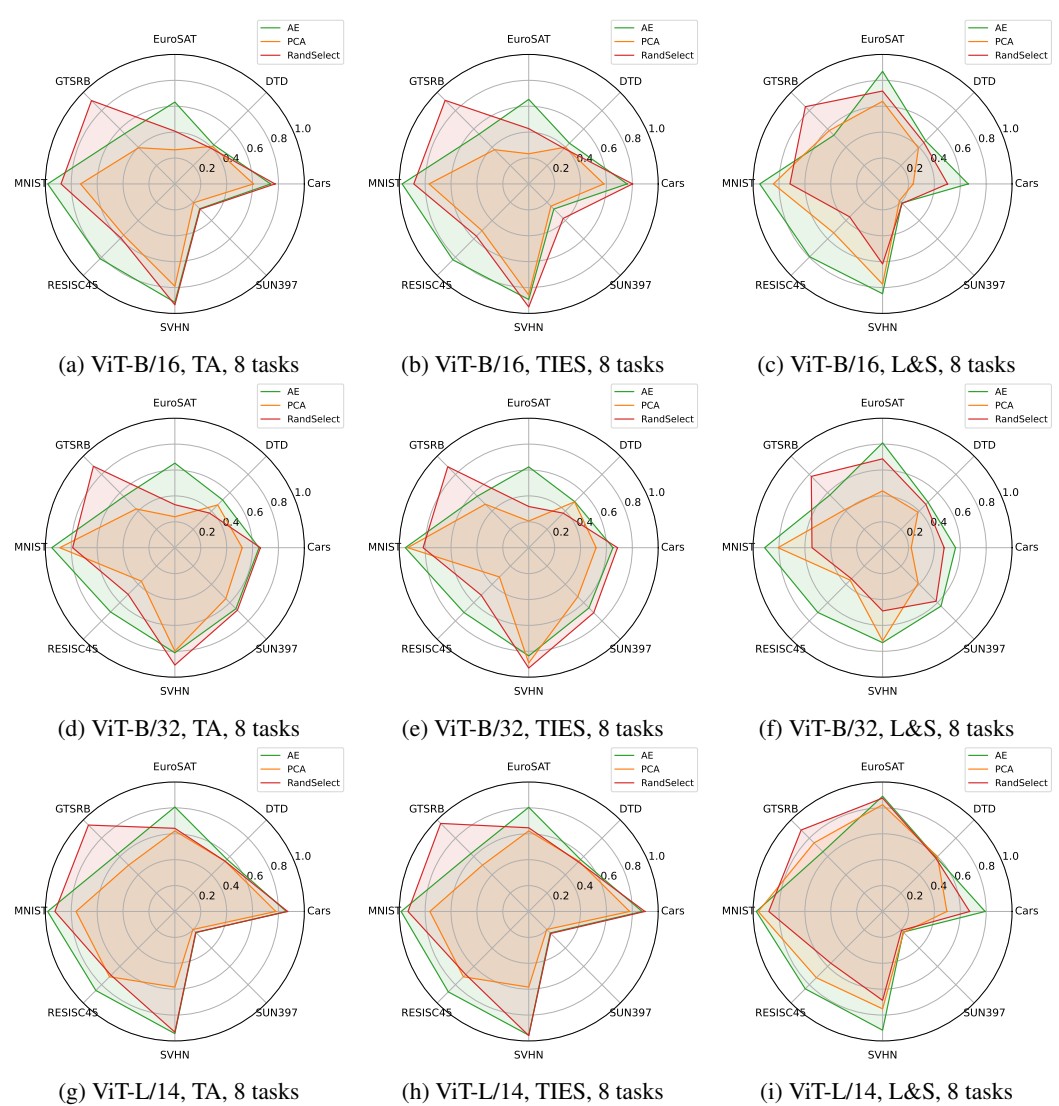

Figure 9: Per dataset bases comparison results for 8 vision tasks of Tab. 1.

## F.3 OFFLINE FEWSHOT OOD GENERALIZATION

**Hyperparameters and Datasets.** For the aTLAS method in the few-shot regime in Sec. 4.1.2, we train for 10 epochs on every target OOD dataset (CIFAR10, CIFAR100, STL10, Food101, Flowers102, OxfordIIITPet). We use the AdamW optimizer with a learning rate of 0.1 and weight decay of 0.1, together with a cosine learning-rate schedule that decays the learning rate from 0.1 to 0 over the course of training. The per-GPU batch size is set to 128 for all models except ViT-L/14, where it is 64 with gradient accumulation of 2, yielding an effective batch size of 128 in both cases.

## F.4 ONLINE CONTINUAL LEARNING

**Hyperparameters and Implementation Details.** In the continual setting, RandSelect maintains a fixed-size buffer of task vectors. As new tasks arrive, their vectors are added to the buffer until it reaches capacity. Once the buffer is full, the method enforces the size constraint by randomly

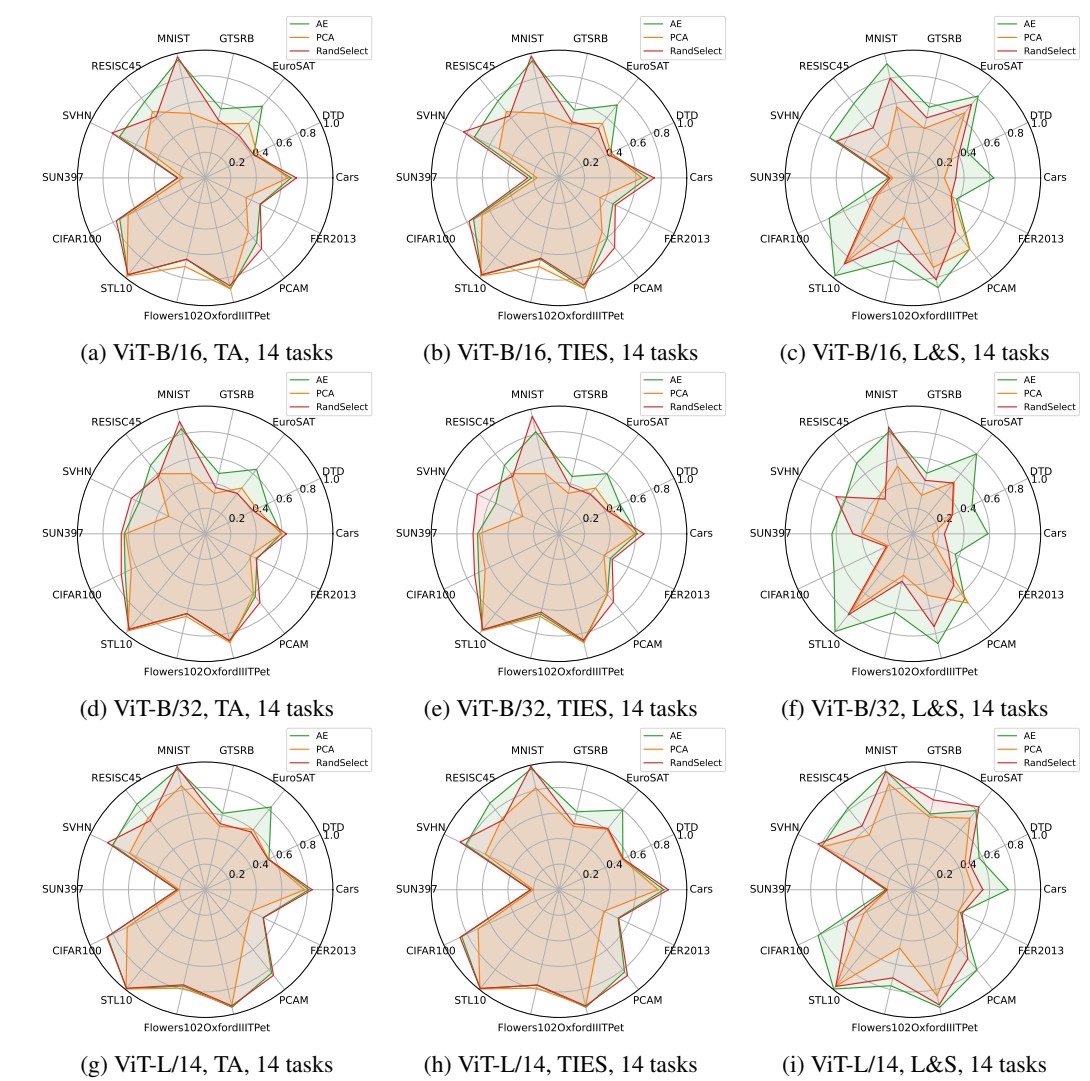

Figure 10: Per dataset bases comparison results for 14 vision tasks of Tab. 1.

Table 16: Online continual addition results (8 tasks, $M = 50\%$) over 5 runs with different task order. We report mean accuracy (%) $\pm$ standard deviation. The bases method with better mean for the same merging method is bold.

| Method | ViT-B/32 | ViT-B/16 | ViT-L/14 |
|---|---|---|---|
| RandSelect-TA | $62.04 \pm 3.69$ | $70.86 \pm 1.95$ | $77.47 \pm 1.40$ |
| AE-TA | $\mathbf{66.60} \pm 0.86$ | $\mathbf{71.23} \pm 2.15$ | $\mathbf{78.99} \pm 1.22$ |
| RandSelect-TSVM | $65.61 \pm 3.15$ | $\mathbf{73.17} \pm 3.82$ | $79.84 \pm 3.24$ |
| AE-TSVM | $\mathbf{69.01} \pm 0.98$ | $73.03 \pm 0.92$ | $\mathbf{80.40} \pm 0.82$ |

discarding one existing task vector whenever a new one is added. For both TA and TSVM, unlike in the offline setting where the isotropic scaling coefficient $\alpha$ is tuned on a validation set, here we fix $\alpha$ to standard values suggested in prior work for simplicity. Specifically, we follow Tang et al. (2025) and set $\alpha = 0.3$ for TA and $\alpha = 1$ for TSVM (Gargiulo et al., 2025), without performing any additional scaling search. Moreover, as shown in Fig. 3, for larger basis sizes ($M = 6, 7$) we observed that using the annealing scheme $\tau = (500, 0.8)$ further improves AE performance;

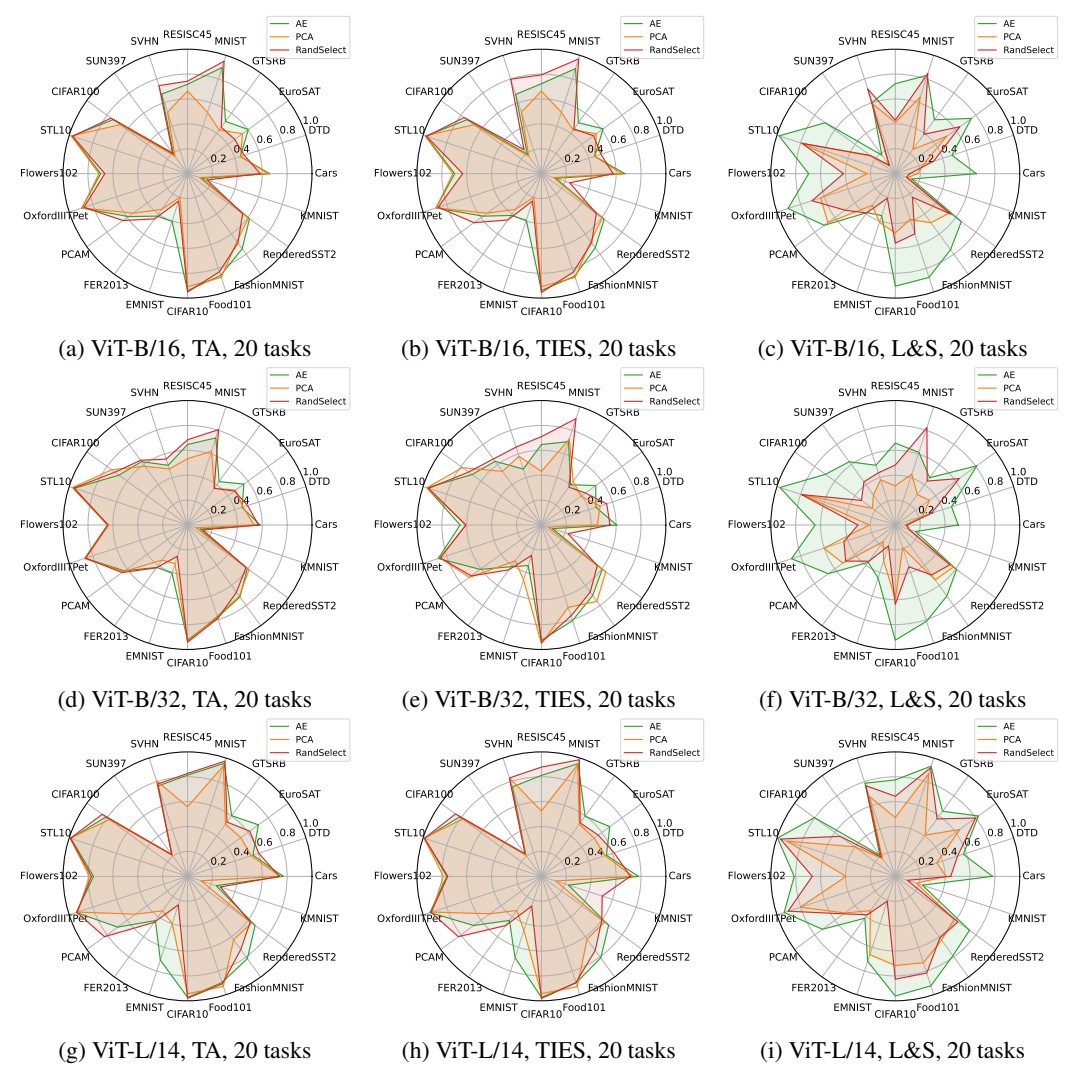

Figure 11: Per dataset bases comparison results for 20 vision tasks of Tab. 1.

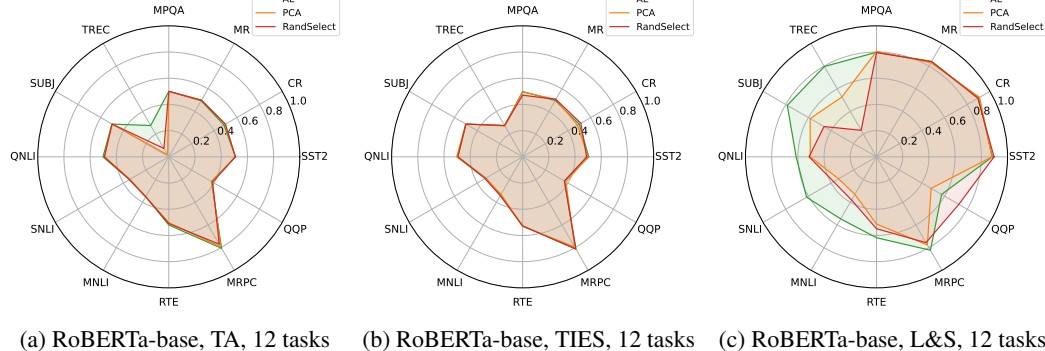

Figure 12: Per dataset bases comparison results for 12 language tasks of Tab. 2.

therefore, we adopt this setting for those runs. For smaller basis sizes ($M < 6$), we continue to use the default hyperparameters for AE basis construction.

**Results across Architectures.** We further include online addition results for different model architectures. From Tab. 16, we see that AE consistently achieves higher mean accuracy with notably lower variance compared to RandSelect across all three backbones. For example, on ViT-B/32, AE improves over RandSelect by more than 4 points under both TA and TSVM merging, while also cutting the standard deviation by a factor of 3–4. Even on larger models such as ViT-L/14, where the margins are smaller, AE still maintains a clear edge in both accuracy and stability. It is worth noting that RandSelect remains a surprisingly strong baseline, particularly in the continual setting. In some cases (e.g., TSVM on ViT-B/16), its mean accuracy approaches that of AE, though at the cost of much higher variance. This suggests that RandSelect can occasionally perform well, but such performance is highly sensitive to task order and thus less reliable. Overall, AE provides a robust and dependable solution for online continual merging, delivering consistently strong results across architectures. The fact that RandSelect can sometimes compete highlights that random bases capture useful task diversity, but also underscores that this phenomenon deserves further investigation in future work.

## F.5 BASES NEGATION

**Hyperparameters and Metrics.** The tuning of $\alpha$ in Tab. 4 is based on selecting the coefficient on the grid search over $[0, 0.05, 0.10, \dots, 1.0]$ (21 grid points) that at least maintain 95% of pretrained model's ImageNet (control task) test accuracy, and we use the selected $\alpha$ to create the edited model and report the target and control metrics.

## G  LLM USAGE STATEMENT

We used LLMs to aid in polishing the writing of this paper. Specifically, LLMs were employed as a general-purpose assistant to improve clarity, grammar, and style, and to suggest alternative phrasings for technical explanations. They were not used to generate novel research ideas, design experiments, or produce results. The authors take full responsibility for all content, including text refined with the assistance of LLMs.

