# OpenReview forum: "Task Vector Bases: A Unified and Scalable Framework for Compressed Task Arithmetic"
_ICLR.cc/2026/Conference — ICLR 2026 Conference Withdrawn Submission_

### Official Review · Reviewer_U1TV · 2025-10-17

**Soundness:** 2
**Presentation:** 3
**Contribution:** 2
**Rating:** 2
**Confidence:** 4

**Summary:**

This paper tackles the challenge of linearly increasing memory and merging costs in task arithmetic as the number of task vectors grows. To overcome this, the authors introduce a compression framework that reduces T task vectors into M task vector bases, while preserving the functional integrity of task arithmetic. The method leverages an autoencoder equipped with softmax activation to ensure that each base vector can be expressed as a convex combination of the original task vectors. Theoretically, the authors prove that arithmetic operations using the compressed vectors retain performance guarantees---up to a constant factor---across addition consistency, out-of-distribution (OOD) generalization, and forgetting resilience. Empirically, the proposed approach achieves task arithmetic performance that matches or exceeds that of random selection and PCA-based compression on a range of vision and NLP tasks.

**Strengths:**

- Practical relevance of the problem formulation: Storing a full set of task vectors for each task in large-scale models is both memory-intensive and computationally expensive during merging. The desire to perform task arithmetic across multiple tasks with a minimal set of vectors reflects a realistic and pressing need in practical deployment settings. The authors' focus on this challenge is timely and well-motivated.

- Limitations of PCA-based compression and proposed remedy: The paper highlights that conventional PCA compression yields task vector bases that are not expressible as non-negative linear combinations of the original task vectors, thereby breaking compatibility with standard task addition. To address this, the authors propose a solution that preserves compositional compatibility.

**Weaknesses:**

- Why autoencoders?: While the authors rightly note that PCA-based compression does not yield basis vectors that can be expressed as non-negative linear combinations of task vectors, the paper does not clarify why alternative dimensionality reduction techniques with inherent non-negativity constraints were not explored. For example, Nonnegative Matrix Factorization (NMF) is a well-established method that decomposes a non-negative matrix into a product of non-negative basis and coefficient matrices. Applied to the task vector matrix $T$, one could decompose $T$ into $T^+$  (where negative entries of $T$ are set to zero) and $T^-$ (where non-negative entries are set to zero) such that $T = T^+ + T^-$, and then apply NMF separately to $T^+$ and $-T^-$. This would enable reconstruction via non-negative combinations, preserving compatibility with conventional task addition. Without a clear justification for favoring autoencoders over such interpretable and constraint-aligned methods, the choice of architecture appears insufficiently motivated.
- Reconstruction in task negation: If my understanding is correct, the authors suggest that task negation is not performed directly in the compressed basis space, but instead involves reconstructing the full set of T task vectors prior to executing the forgetting operation. If such reconstruction is deemed acceptable in the context of negation, it raises a key question: why isn’t the same reconstruction strategy employed for task addition? In fact, if reconstruction is permitted, one could simply store PCA-compressed task vectors and reconstruct them at merge time to apply conventional task arithmetic techniques. The paper does not address this inconsistency, leaving a significant conceptual gap in the proposed framework.
- Experimental configuration of M: In the experiments, the number of compressed bases M is set as a fixed proportion (e.g., 50% or 25%) of the total number of task vectors T. However, in realistic applications where computational and storage resources are limited, it is far more common for M to be fixed independently of T. This makes the experimental design less reflective of practical deployment conditions. Providing results under a fixed M setting regardless of T would offer a more robust and practically relevant evaluation. Demonstrating effectiveness under such constraints would significantly strengthen the credibility of the proposed method.
- Competition with Parameter-Efficient Fine-Tuning (PEFT): While task arithmetic traditionally involves full-parameter task vectors, recent advances increasingly leverage PEFT methods such as LoRA[1], which offer significant improvements in memory efficiency and merging cost. These approaches may not only be more scalable than full-parameter models but could also outperform the proposed base-compression strategy in both efficiency and task performance. Despite this, the paper lacks a comprehensive comparison between the proposed method and PEFT-based alternatives in terms of memory footprint, merging complexity, and empirical performance.

[1] Zhang, Jinghan, Junteng Liu, and Junxian He. "Composing parameter-efficient modules with arithmetic operation." Advances in Neural Information Processing Systems 36 (2023): 12589-12610.

**Questions:**

See weaknesses

---

### Official Review · Reviewer_PS3P · 2025-10-20

**Soundness:** 2
**Presentation:** 3
**Contribution:** 3
**Rating:** 4
**Confidence:** 4

**Summary:**

The paper advances Task Arithmetic (Ilharco et al., 2023) by proposing an orthogonal framework aimed at reducing storage and computational costs of task vectors. For achieving their goal, the authors propose to learn a basis of $M \leq T$ from a set of the original $T$ task vectors, using a linear autoencoder with softmax activation. The authors provide both empirical and theoretical justifications to support their design choices, offering bounds on error incurred from compression. The proposed framwork is tested on offline and online Task Arithmetic test beds on both vision and language domains.

**Strengths:**

- The positioning of the paper in literature is clear, offering a framework that tackles a novel problem.
- The method is compelling, as it is solidly grounded in theory, allowing for using classical results from spectral analysis.
- Considering the constraints of the problem, the experimental validation is extensive (also, it is very useful that the authors compared with simple PCA) and the results are promising.
- The paper is well-written, the figures/plots help the narrative and understanding of the results. Finally, the mathematical notation is clear.

**Weaknesses:**

Overall, the paper is solid. However, in light of recent advances in Task Arithmetic, the following points require some attention:

**W1.**

[1,2,3,4] prove and support the idea that linearization around the pre-trained parameters is the key enabler of proper Task Arithmetic. The intuition is that, when task vectors implement functions that are linear w.r.t. the weights of the model, the composed model will act as a linear combination of orthogonal functions (thus, allowing for minimal interference across task vectors eg. see Fig. 2 of [4] or Fig. 3 of [3]).

Being these vectors orthogonal by design (when tasks are statistically different), they should already form a basis by themselves, without any need for the proposed methodology: compressing linearized task vectors, intuitively, simply reduces to discarding some task vectors that, possibly due to tasks being very similar (in a statistical sense), will approximately point in the same direction in parameter space.

In essence, it could be interesting to observe what happens when compressing task vectors obtained from linearized fine-tuning [1].

-------------------

**W2.**

I'm finding Section 3.2.3 weakly linked to the rest of the paper: up until the end of Section 3.2.1 the narrative focuses on basis arithmetic, reconstruction errors of your framework etc. Then, abruptly the paper presents some results regarding multi-task/OOD loss bounds. I'd suggest the authors to strengthen the connection with the previous section and why these bounds matter.

To be honest, I'm not even sure the theorems/bounds of this section are needed for the narrative of your paper, as they partially overlap with already established literature which proves slightly stronger results (eg. see [5], which also characterizes the second-order term of the Taylor expansion, or [6] which characterizes task addition, negation and OOD generalization with non-linear transformers under the light of task alignment). I'd suggest to either remove this part, or clearly highlight the contribution and novelty of your theoretical results.

-------------------

**Minor Weaknesses.**
- I would reframe your contributions: the first three read as a single contribution (i.e. your framework).
- Before/After Theorem 3.4 (LL204-206), I'd suggest to add a pointer to the proof.

-------------------

**_References:_**

[1] Ortiz-Jimenez, Guillermo, Alessandro Favero, and Pascal Frossard. "Task arithmetic in the tangent space: Improved editing of pre-trained models." NeurIPS 2023.

[2] Jin, Ruochen, et al. "Fine-tuning attention modules only: Enhancing weight disentanglement in task arithmetic." ICLR 2025.

[3] Iurada, Leonardo, Marco Ciccone, and Tatiana Tommasi. "Efficient model editing with task-localized sparse fine-tuning." ICLR 2025.

[4] Tang, Anke, et al. "Parameter efficient multi-task model fusion with partial linearization." ICLR 2024.

[5] Porrello, Angelo, et al. "A second-order perspective on model compositionality and incremental learning." ICLR 2025.

[6] Li, Hongkang, et al. "When is task vector provably effective for model editing? a generalization analysis of nonlinear transformers." ICLR 2025.

**Questions:**

Thanking in advance for their response, I'd kindly invite the authors to address the points raised in the Weaknesses section of this review.

---

### Official Review · Reviewer_yqA6 · 2025-11-01

**Soundness:** 2
**Presentation:** 3
**Contribution:** 2
**Rating:** 4
**Confidence:** 3

**Summary:**

The paper introduces a task-vector–basis framework that compresses a large set of task vectors (T) into a small set of basis vectors (M) while attempting to preserve “task arithmetic” (addition/subtraction) semantics. The core mechanism is a linear autoencoder whose coefficient layer uses a softmax to produce simplex-constrained combinations, encouraging faithful reconstruction of the original task-vector geometry. The authors argue that PCA distorts the functional directions underlying task arithmetic and motivate their design with a geometric analysis. Empirically, the method shows advantages over simple baselines at higher compression ratios and evaluates both additive and subtractive compositions, in partial agreement with the theory.

**Strengths:**

1. Clear diagnosis of PCA’s limitations. The geometric argument for why PCA fails to preserve task-arithmetic semantics is insightful and well-motivated, and it naturally motivates the proposed approach.
2. Solid results at higher compression. When compressing a larger collection of task vectors, the method outperforms simple baselines; these observations align with the authors’ theoretical analysis.
3. Covers both addition and subtraction. Evaluating both additive and subtractive task arithmetic offers a more complete view of functionality preservation.

**Weaknesses:**

1. Questionable practical motivation for very large T. A major stated goal is reducing storage overhead for many task vectors. Yet in practice, the quality of simple task-vector merging typically degrades rapidly as T grows; much prior work evaluates in the ~8–20 range. If real deployments rarely maintain very large libraries of task vectors due to performance collapse, the storage-savings motivation is less compelling without concrete use cases.
2. Limited baselines. The main experiments compare against only three Task-Vector variants. Recent, stronger baselines—e.g., Parameter Competition Balancing for Model Merging [1], Task Arithmetic in the Tangent Space [2], and Beyond Task Vectors: Selective Task Arithmetic [3]—report substantially better results than the original Task Vector. Omitting them weakens the empirical claim.
3. Architecture sensitivity is unclear. The method relies on a softmax-activated linear autoencoder to form bases. There is no ablation on the autoencoder design (e.g., softmax vs. simplex constraints via projection, temperature, coefficient sparsity, nonlinearity), so robustness to architectural choices is unknown.
4. Compute/storage savings not quantified. The paper lacks concrete measurements of wall-clock speed, memory footprint, and I/O savings attributable to compression. Without numbers (for, say, T=100 and T=1000), it is difficult to evaluate the cost-effectiveness of the approach.
5. Non-trivial performance loss on many datasets. In several cases, the compressed basis representation trails the original Task Vector approach. The paper should analyze whether the accuracy drop is acceptable relative to storage/computation gains, ideally with a Pareto plot and quantified trade-offs.

References:
[1] Parameter Competition Balancing for Model Merging.
[2] Task Arithmetic in the Tangent Space: Improved Editing of Pre-Trained Models.
[3] Beyond Task Vectors: Selective Task Arithmetic Based on Importance Metrics.

**Questions:**

1. What concrete deployment scenarios require large T where storage dominates yet task arithmetic remains useful?
2. How does the method compare to stronger baselines under their setups?
3. How sensitive are results to the autoencoder design (softmax temperature, sparsity, projection)?
4. What are the exact storage/compute savings and the accuracy–compression Pareto curves?
5. Where does it fail most, and why (e.g., under-represented directions, bias toward majority tasks)?

f the authors satisfactorily address these issues, I would raise my score.

---

### Author Response · Authors · 2025-11-19
**For Future Readers**

We want to thank all reviewers for questions and comments. After our discussion, we plan to withdraw our paper, but we still want to clarify a few concerns here for future readers:

1. Motivation (yqA6): There are many practical scenarios (supported by U1TV) where we do need to constrain ourselves to a fixed number of task bases, given that:
    * Even for 8-20 tasks, checkpoint storage and GPU memory cost (line 41-45) for LLMs (Sec F.2) are both nontrivial.
    * For an online setting (line 92, algorithm 1, Sec 4.1.3), $T$ can grow to infinity and it’s simply impossible to have access to all task vectors in the stream.
2. Why theory (PS3P): Note that under the framework of Sec 3.2.3, we can **compare the performance of our bases methods and original task arithmetic methods theoretically**. As yqA6 pointed out, our empirical observations later on in Sec.4 (where subsections exactly relate to each of our theorems) align with our theorems. Therefore, Sec 3.2.3 is one of the novel contributions that suits well with the context of this paper.
3. “Why autoencoders” (U1TV), not Nonnegative Matrix Factorization: We’ve explained the detailed reasons in Sec. A.4. In what follows we provide a high-level summary: For the method that U1TV described, line 924-929 elaborated the details, and roughly speaking, the advantage of our method is **softmax**ed-AE where softmax activation brings
    * First, the bases-merged model (See Fig.1b) still lives in the original task addition cone, thus not losing too much information after compression.
    * Second, allows us to guarantee that (line 178) “each basis vector can be interpreted as a **convex combination of input task vectors**,” so in Theorem 3.5 & 3.6 we only see almost no penalty with softmax-AE bases for addition and OOD generalization from compression. Nonnegativity itself is not sufficient, and AE is a basic framework where we can insert additional constraints. **For next steps, we welcome the community to think of data-aware compression methods built on top of our work.**
4. Why reconstruction in task negation but not in addition (U1TV): (line 107-108) All task vectors must be accessible during task negations (as you need to forget each target task), yet for addition since we only need the multitask capability, maintaining $T$ task is not necessarily needed and could be significantly improved with bases type of methods. Task vector compression is extremely important for online settings as well.
5. Experiments:
- Architecture sensitivity (yqA6): For sensitivity of temperature ($\approx$ sparsity), see Sec E.1.
- Compute/Storage savings against Accuracy (yqA6): See Figure 2c for results on standard model merging benchmarks.
- Baselines:
    * Linearized finetuning (PS3P, yqA6): Note that even in our softmax-AE bases, we do not restrict ourselves to a set of orthogonal bases. More importantly, orthogonality-enforced bases, such as PCA, as we illustrated both conceptually and empirically, are not sufficient to guarantee good task arithmetic performance. We expect the relationship between RandSelect, PCA, and AE will be the same on top of the linearly finetuned task vectors.
    * PEFT (U1TV): This is similar to sparsity and quantization line of work (line 465-470) where compression is acted along the dimension of $d$ and our work focuses on compressing the dimension of $T$. That said, first, for the online setting (line 92, algorithm 1, Sec 4.1.3) when $T \to \infty$, we have to constrain ourselves to a fixed number of bases, thus making comparison with $d$-dim methods unfair. Second, for the offline setting, we can test bases on top of all these $d$-dim methods for further compression.
    * Other mentioned baselines (yqA6): In Sec. 4, we’ve included TA, TIES, L&S, aTLAS, TSVM, in total 5 methods as the test bed for all bases methods comparison. Note that the last three are more recent methods and L&S achieved the best overall results in a recent LLM model merging benchmark [1]. On standard 8 task vision benchmarks, L&S, aTLAS, TSVM all achieve better performance than the papers [2,3] mentioned by yqA6. [4] is not directly comparable because [4] is evaluated on a 6-task vision benchmark.

[1] He, Yifei, et al. "MergeBench: A Benchmark for Merging Domain-Specialized LLMs." The 39th NeurIPS Datasets and Benchmarks Track (2025).

[2] Ortiz-Jimenez, Guillermo, Alessandro Favero, and Pascal Frossard. "Task arithmetic in the tangent space: Improved editing of pre-trained models." Advances in Neural Information Processing Systems 36 (2023): 66727-66754.

[3] Du, Guodong, et al. "Parameter competition balancing for model merging." Advances in Neural Information Processing Systems 37 (2024): 84746-84776.

[4] Bowen, Tian, et al. "Beyond task vectors: Selective task arithmetic based on importance metrics." arXiv preprint arXiv:2411.16139 (2024).

---

### Note · Authors · 2025-12-02

I have read and agree with the venue's withdrawal policy on behalf of myself and my co-authors.